



# An improved second order dynamic stall model for wind turbine airfoils

Galih Bangga[1], Thorsten Lutz[1], and Matthias Arnold[2]

[1]Institute of Aerodynamics and Gas Dynamics (IAG), University of Stuttgart, 70569 Stuttgart, Germany
[2]Wobben Research and Development GmbH, 26607 Aurich, Germany

**Correspondence:** Galih Bangga (bangga@iag.uni-stuttgart.de)

**Abstract.** Robust and accurate dynamic stall modeling remains one of the most difficult tasks in wind turbine load calculations despite its long research effort in the past. In the present paper, a new second order dynamic stall model is developed with the main aim to model the higher harmonics of the vortex shedding while retaining its robustness for various flow conditions and airfoils. Comprehensive investigations and tests are performed by varying many flow parameters. The occurring physical characteristics for each case are discussed and evaluated in the present studies. The improved model is also tested on four different airfoils with different relative thicknesses. The validation against measurement data demonstrates that the improved model is able to reproduce the dynamic polar accurately without airfoil specific parameter calibration for each investigated flow condition and airfoil. This can deliver further benefit to industrial applications where experimental/reference data for calibrating the model is not always available.

## 1 Introduction

An accurate prediction of wind turbine blade loads is influenced by many parameters including 3D and unsteady effects. The first mainly occurs in the root and tip areas of the blade due to radial flow and induced velocity influences, respectively (Bangga, 2018). The latter can occur due to variation of the inflow conditions caused by yaw misalignment, wind turbulence, shear & gusts, tower shadow and aeroelastic effects of the blade. The above mentioned phenomena may result in dynamic stall (DS). Experimental studies (Martin et al., 1974; Carr et al., 1977; McAlister et al., 1978) showed that the aerodynamic forces can differ significantly in comparison to the static condition. DS is often initiated by the generation of a leading edge vortex (LEV), which increases positive circulation effect on the airfoil suction side causing delayed stall. This intense leading edge vortex is convected downstream along the airfoil towards the trailing edge. At the same time, the lift force increases significantly and the pitching moment becomes more negative compared to the static values. A significant drag increases is observed at large angles of attack. An example is shown in Figure 1. Afterwards, a trailing edge vortex (TEV) with opposite rotational direction than LEV is formed, which pushes the leading edge vortex towards the wake area. This onset may result in a significant drop of the lift coefficient ($C_L$) and can be dangerous for the blade structure itself.

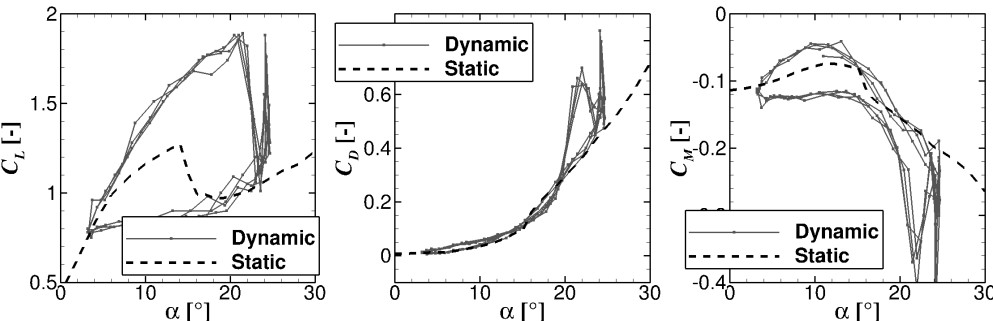

**Figure 1.** Typical dynamic stall behavior of S801 airfoil. Data obtained from (Ramsay et al., 1996).

To model the behavior of the airfoil under these situations, semi-empirical models can be used. The models are known to

produce reasonable results with very small computational effort. Leishman & Beddoes (LB) (Leishman and Beddoes, 1989) have developed a model for dynamic stall combining the flow delay effects of attached flow with an approximate representation of the development and effect of separation (Larsen et al., 2007). This model was developed for helicopter applications and therefore includes a fairly elaborate representation of the nonstationary attached flow depending on the Mach number and a rather complex structure of the equations representing the time delays (Larsen et al., 2007). Hansen et al. (Hansen et al., 2004)

simplified the model for wind turbine applications by removing the consideration of compressibility effects and the leading edge separation. The latter was argued because the relative thickness of wind turbine airfoil is typically no less than 15%. This model was called Risø model in (Larsen et al., 2007). Examples of the other models are given by Øye (Øye, 1991), Tran & Petot (ONERA model) (Tran and Petot, 1980) and Tarzanin (Boeing-Vertol model) (Tarzanin, 1972). To better model the vortex shedding characteristics at large angles of attack, second order dynamic stall models were introduced. An example of

this model was given by Snel (Snel, 1997) which makes use of the difference between the inviscid to the viscous static polar data as a main forcing term for the dynamic polar reconstruction, in contrast to the LB model that uses the changes of the angle of attack over the time. An improved version of the Snel model was proposed recently by Adema (Adema et al., 2019) to cover for the increased shedding effects in the downstroke phase. All above mentioned models employ the static polar data and dynamic flow parameters as the input needed for the dynamic polar reconstruction. Then, the models compute the dynamic

force difference required for the reconstruction process.

Although many attempts have been dedicated for dynamic stall modeling (Gupta and Leishman, 2006; Larsen et al., 2007; Adema et al., 2019; Elgammi and Sant, 2016; Wang and Zhao, 2015; Sheng et al., 2006; Galbraith, 2007; Sheng et al., 2008), engineering calculations in industry are still relying on the very basic classical dynamic stall models such as the Leishman-Beddoes and Snel models. The reason is the simplicity to tune in the models for different airfoils and for different flow

conditions. Therefore, one major key for a model to be used in industrial applications is robustness of the model itself. The main purpose of this paper is to document widely used state-of-the-art dynamic stall models in research and industries. These include the first order LB model and the second order Snel model as well as the ONERA model. A very recently improved



Snel model according to Adema (Adema et al., 2019) will also be evaluated. The mathematical formulations of these model will be presented in this report. Weaknesses of existing dynamic stall modeling shall be identified, and possible corrections to those limitations will be described. Finally, a new second order dynamic stall modeling will be proposed that is able to model not only the second order lift and drag forces, but also the pitching moment along with calculation examples in comparison to experimental data for different airfoils and flow conditions.

The paper is organized as following. Section 2 describes the mathematical formulation of four state-of-the-art dynamic stall models and the new model developed in this work. Then, in Section 3 assessments are carried out on the sensitivity of each model to time step variation and how each model performs in comparison with measurement data. The new model is further tested at various flow conditions, and to examine its robustness on four different airfoils without further calibrating the constants. Finally, all results will be concluded in Section 4.

## 2 Mathematical Formulations

In this section the mathematical formulations of each model are described in detail. The reasons are manly to provide information on how each model was employed and to gain deeper insights for further developing the new model. Note that each existing model was developed by different authors, thus different symbols and formulation methods were adopted in those publications (Beddoes, 1982; Leishman, 1988; Leishman and Beddoes, 1989; Tran and Petot, 1980; Dat and Tran, 1981; Petot, 1989; Snel, 1997; Adema et al., 2019). In this paper, all models are described in a consistent way for clarity and for easier interpretation/implementation process.

### 2.1 Leishman-Beddoes model

The original Leishman-Beddoes model is composed by three main contributions representing various flow regimes: (1) unsteady attached flow, (2) unsteady separated flow and (3) dynamic stall. The present section will elaborate the mathematical description and its physical interpretation of each module. Figure 2 illustrates several main parameters needed for modeling the dynamic stall characteristics.

### 2.1.1 Unsteady attached flow

In this module, the unsteady aerodynamic response of the loads is represented by the time delay effects. The indical formulas were constructed based on the work of Beddoes (Beddoes, 1982), and have been refined by Leishman (Leishman, 1988). The loads are assumed to originate from two main sources; one for the initial noncirculatory loading from the piston theory and another for the circulatory loading which builds up quickly to the steady state value (Leishman and Beddoes, 1989). In the formulation, the relative distance traveled by the airfoil in terms of semi-chords is represented by $s = 2Vt/c$ that can be used also to describe the nondimensional time. Note that $V$, $t$ and $c$ are freestream wind speed, time and chord length, respectively. For a continuously changing angle of attack $\alpha_n$, the effective angle of attack ($\alpha_{e_n}$) can be represented as:

$$\alpha_{e_n} = \alpha_n - X_n - Y_n \tag{1}$$





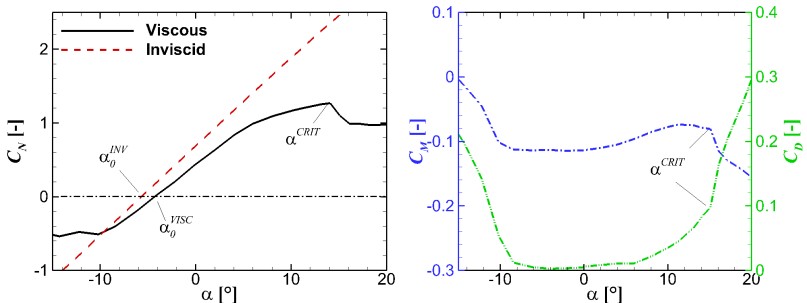

**Figure 2.** Illustration of main aerodynamic parameters needed for modeling the dynamic stall characteristics.

where $n$ is the current sample time. The last two terms describe the deficiency functions that are given by:

$$X_n = X_{n-1} \exp\left(-b_1\beta^2\Delta s\right) + A_1 \, \Delta\alpha_n \exp\left(-b_1\beta^2\Delta s/2\right) \tag{2}$$

$$Y_n = Y_{n-1} \exp\left(-b_2\beta^2\Delta s\right) + A_2 \, \Delta\alpha_n \exp\left(-b_2\beta^2\Delta s/2\right) \tag{3}$$

where

$$\Delta\alpha_n = \alpha_{n+1} - \alpha_n \tag{4}$$

$$\Delta s = s_n - s_{n-1}. \tag{5}$$

In these equations, $b_1$, $b_2$, $A_1$ and $A_2$ are constants. The variable $\beta$ represents the compressibility effects and is formulated as $\beta = \sqrt{1 - M^2}$. Because information about the previous cycle is needed in the formulations, initializations are required. The solution needs to develop for a certain time until convergence of the resulting unsteady loads is obtained.

The circulatory normal force due to an accumulating series of step inputs in angle of attack can be obtained using

$$C_{N_n}^C = \frac{dC_N}{d\alpha}\left(\alpha_{e_n} - \alpha_0^{INV}\right) \tag{6}$$

The variable $\alpha_{0_{INV}}$ is the angle of attack for zero inviscid normal force. The original formulation of the model disregarded the use of $\alpha_0^{INV}$. However, this term is important when the airfoil has a finite camber. This has been pointed out as well by Hansen et al. (Hansen et al., 2004).

The noncirculatory (impulsive) normal force is obtained by

$$C_{N_n}^I = \frac{4K_\alpha T_I}{M}\left(\frac{\Delta\alpha_n}{\Delta t} - D_n\right). \tag{7}$$

where $T_I$ is given by $T_I = Mc/V$. The deficiency function $D_n$ is given by

$$D_n = D_{n-1} \exp\left(\frac{-\Delta t}{K_\alpha T_I}\right) + \left(\frac{\Delta\alpha_n - \Delta\alpha_{n-1}}{\Delta t}\right) \exp\left(\frac{-\Delta t}{2K_\alpha T_I}\right), \tag{8}$$

and $\Delta t = t_n - t_{n-1}$.



The total normal force coefficient under attached flow conditions is given by the sum of circulatory and noncirculatory components as

$$C_{N_n}^P = C_{N_n}^C + C_{N_n}^I. \tag{9}$$

### 2.1.2 Unsteady separated flow

Leishman & Beddoes (Leishman and Beddoes, 1989) stated that the onset of leading edge separation is the most important aspect in dynamic stall modeling. The condition at when leading edge stall occurs, is controlled by a critical leading edge pressure coefficient that is linked into the formulation by defining a lagged normal force coefficient $C_{N_n}^{P1}$ as:

$$C_{N_n}^{P1} = C_{N_n}^P - D_{p_n} \tag{10}$$

where $D_{p_n}$ is given by

$$D_{p_n} = D_{p_{n-1}} \exp\left(-\frac{\Delta s}{T_p}\right) + \left(C_{N_n}^P - C_{N_{n-1}}^P\right) \exp\left(-\frac{\Delta s}{2T_p}\right). \tag{11}$$

It has been investigated by Leishman & Beddoes (Leishman and Beddoes, 1989) that the calibration time constant $T_p$ is largely independent of the airfoil shape. The substitute value of the effective angle of attack incorporating the leading edge pressure lag response may be obtained using

$$\alpha_{f_n} = \alpha_0^{INV} + \left(\frac{C_{N_n}^{P1}}{dC_N/d\alpha}\right) \tag{12}$$

In most of airfoil shapes, the progressive trailing edge separation causes loss of circulation and introduces nonlinear effects on the lift, drag and pitching moment, especially on cambered airfoils. This is even more important for wind turbine airfoils because the relative thickness is large. To derive a correlation between the normal force coefficient with the separation location ($f_n$), the relation based on the flat plate from Kirchhoff/Helmholtz can be used, that reads:

$$C_{N_n}^{VISC} = \frac{dC_N}{d\alpha} \left(\frac{1+\sqrt{f_n}}{2}\right)^2 (\alpha_n - \alpha_0^{VISC}). \tag{13}$$

Note that $\frac{dC_N}{d\alpha}$ is the inviscid polar gradient determined in Section 2.6.

The location of the separation point is usually obtained by a curve-fitting procedure in literature. For example, Leishman & Beddoes (Leishman and Beddoes, 1989) proposed the following correlation

$$f_n = \begin{cases} 1 - 0.3 \exp\left(\dfrac{\alpha_n - \alpha_1}{S_1}\right); & \alpha_{f_n} \leq \alpha_1 \\ 0.04 + 0.66 \exp\left(\dfrac{\alpha_1 - \alpha_n}{S_2}\right); & \alpha_{f_n} > \alpha_1 \end{cases} \tag{14}$$

The coefficients $S_1$ and $S_2$ define the static stall characteristic while $\alpha_1$ defines the static stall angle. The derivation was based on the NACA 0012, HH-02 and SC-1095 airfoils that have a single break point of the static lift force coefficient. Gupta &





Leishman (Gupta and Leishman, 2006) proposed the formulation for the S809 airfoil as:

$$
f_n = \begin{cases}
c_1 + a_1 \exp(S_1 \alpha_n); & \alpha_{f_n} \leq \alpha_1 \\
c_2 + a_2 \exp(S_2 \alpha_n); & \alpha_1 < \alpha_{f_n} < \alpha_2 \\
c_3 + a_3 \exp(S_3 \alpha_n): & \alpha_{f_n} \geq \alpha_2
\end{cases}
\tag{15}
$$

that has two break points ($\alpha_1$ and $\alpha_2$) of the static lift force coefficient, where $c_1$, $c_2$, $c_3$, $a_1$, $a_2$ and $a_3$ are constants.

The additional effects of the unsteady boundary layer response may be represented by application of a first-order lag to the value of $f_n$ to produce the final value for the unsteady trailing edge separation point $f_{2_n}$ (Leishman and Beddoes, 1989). This can be represented as

$$
f_{2_n} = f_n - D_{f_n}
\tag{16}
$$

where $D_{f_n}$ is given by

$$
D_{f_n} = D_{f_{n-1}} \exp\left(-\frac{\Delta s}{T_f}\right) + (f_n - f_{n-1}) \exp\left(-\frac{\Delta s}{2T_f}\right),
\tag{17}
$$

and $T_f$ is a constant. Then, the unsteady viscous normal force coefficient for each sample time can be obtained using

$$
C_{N_n}^f = \frac{dC_N}{d\alpha} \left(\frac{1 + \sqrt{f_{2_n}}}{2}\right)^2 (\alpha_{e_n} - \alpha_0^{VISC}) + C_{N_n}^I
\tag{18}
$$

The tangential component of the force can be obtained by (Leishman and Beddoes, 1989):

$$
C_{T_n}^f = -\eta \frac{dC_N}{d\alpha} \alpha_{e_n}^2 \sqrt{f_{2_n}}
\tag{19}
$$

note that positive $C_{T_n}^f$ is defined in the direction of the trailing edge while $\eta$ is a constant.

According to Leishman & Beddoes (Leishman and Beddoes, 1989) and Gupta & Leishman (Gupta and Leishman, 2006), a general expression for the pitching moment behavior cannot be obtained from Kirchhoff theory, and an alternative empirical relation must be formulated. Gupta & Leishman (Gupta and Leishman, 2006) proposed the following formulation for the S809

airfoil

$$
C_M^f = \begin{cases}
C_{M_0} + \left(K_0 + K_1(1 - f_{2_n}) + K_2 \sin(\pi f_{2_n}^m)\right); & \alpha_n \leq \alpha_2 \\
C_{M_0} + \left(K_0 + K_3 \exp(K_4 f_{2_n}^m)\right); & \alpha_n > \alpha_2
\end{cases}
\tag{20}
$$

where $C_{M_0}$ defines the moment coefficient at zero normal force and $K_0$ is the mean offset of the aerodynamic center from the quarter chord position, $K_1$, $K_2$, $K_3$, $K_4$ and $m$ are constants.

### 2.1.3    Dynamic stall

The third part of the model describes the post-stall characteristics where the vortical disturbances near the leading edge become stronger. The effect of vortex shedding is given by defining the vortex lift as the difference between the linearized value of the



unsteady circulatory normal force and the unsteady nonlinear normal force obtained from the Kirchhoff approximation, that reads

$$C_{V_n} = C_{N_n}^C (1 - K_n) \tag{21}$$

where $K_n$ is given by

$$K_n = \frac{1}{4} \left( 1 + \sqrt{f_{2_n}} \right)^2. \tag{22}$$

The normal force is allowed to decay, but it is updated with a new increment in the normal force based on prior forcing condition, that can be defined as

$$C_{N_n}^V = \begin{cases} C_{N_{n-1}}^V \exp\left(-\dfrac{\Delta s}{T_v}\right) + \left(C_{V_n} - C_{V_{n-1}}\right) \exp\left(-\dfrac{\Delta s}{2T_v}\right); & \text{if} \quad 0 < \tau_{v_n} < T_{vl} \\ C_{N_{n-1}}^V \exp\left(-\dfrac{\Delta s}{T_v}\right); & \text{otherwise} \end{cases} \tag{23}$$

where $T_v$ and $T_{vl}$ are the vortex decay and center of pressure travel time constants, respectively. The nondimensional vortex time is given by (dos Santos Pereira, 2010; Elgammi and Sant, 2016):

$$\tau_{v_n} = \begin{cases} \tau_{v_{n-1}} + 0.45 \dfrac{\Delta t}{c} V; & \text{if} \quad C_{N_n}^{P1} > C_N^{CRIT} \\ 0; & \text{if} \quad C_{N_n}^{P1} < C_N^{CRIT} \quad \text{and} \quad \Delta \alpha_n > 0 \end{cases} \tag{24}$$

with $C_N^{CRIT}$ being the inviscid critical static normal force, usually indicated by the break of the (viscous) moment polar at the critical angle of attack $\alpha_n^{CRIT}$. This can be formulated as:

$$C_N^{CRIT} = \frac{dC_N}{d\alpha} (\alpha_n^{CRIT} - \alpha_0^{INV}). \tag{25}$$

The idealized variation of the center of pressure with the convection of the leading edge vortex can be modeled by

$$C_{Pv_n} = K_v \left( 1 - \cos\left(\frac{\pi \tau_v}{T_{vl}}\right) \right) \tag{26}$$

The dynamic moment coefficient can be formulated as

$$C_{M_n}^V = -C_{Pv_n} C_{N_n}^V \tag{27}$$

Therefore, the total dynamic loading on the airfoil from all modules can be written as

$$C_{N_n}^D = C_{N_n}^f + C_{N_n}^V \tag{28}$$
$$C_{T_n}^D = C_{T_n}^f \tag{29}$$
$$C_{M_n}^D = C_{M_n}^f + C_{M_n}^V \tag{30}$$

and by converting these forces into lift and drag, one obtains:

$$C_{L_n}^D = C_{N_n}^D \cos \alpha_n - C_{T_n}^D \sin \alpha_n \tag{31}$$
$$C_{D_n}^D = C_{N_n}^D \sin \alpha_n + C_{T_n}^D \cos \alpha_n \tag{32}$$



### 2.1.4 Note to present implementation

In Equations (14) and (15), a curve-fitting procedure is usually adopted in literature. In this sense, the parameters or even the formulation need to be adjusted when the airfoil is different. Therefore, in the present implementation, the separation point is derived directly from the static polar data using inversion of Equation (13) as.

$$f_n = \left( 2 \sqrt{\frac{C_{N_n}^{VISC}}{\frac{dC_N}{d\alpha}(\alpha_{f_n} - \alpha_0^{VISC})}} - 1.0 \right)^2 \tag{33}$$

The same approach was used for example by Hansen et al. (Hansen et al., 2004). This way, the user can avoid dealing with curve fitting adjustment (which requires changes on the constants for different airfoils and flow conditions) as long as the static polar data is available.

In the original formulation, the pitching moment is obtained also by a curve fitting procedure in Equation (20). Again, this kind of approach is not straightforward as the user needs to perform curve fitting of the polar data. In the present implementation, the moment coefficient is easily obtained from the static viscous polar data by interpolating the value at the effective angle of attack incorporating the leading edge pressure time lag $\alpha_{f_n}$, that reads

$$C_{M_n}^f = C_M^{VISC}(\alpha_{f_n}). \tag{34}$$

In this sense, the moment coefficient can be reconstructed easily without the need to adjust the parameters in advance, minimizing the user error.

Furthermore, to avoid discontinuity in the downstroke phase for Equation (24), an additional condition is applied in the present implementation as:

$$\tau_{v_n} = \begin{cases} \tau_{v_{n-1}} + 0.45 \dfrac{\Delta t}{c} V; & \text{if} \quad C_{N_n}^{P1} > C_N^{CRIT} \\ 0; & \text{if} \quad C_{N_n}^{P1} < C_N^{CRIT} \quad \text{and} \quad \Delta\alpha_n \geq 0 \\ \tau_{v_{n-1}}; & \text{otherwise} \end{cases} \tag{35}$$

### 2.2 ONERA model

The ONERA dynamic stall model was originally developed by Tran and Petot (Tran and Petot, 1980; Dat and Tran, 1981; Petot, 1989). The model is constructed by two non-linear differential equations describing the characteristics of the dynamic lift coefficient. The first equation defines the inviscid response of the airfoil, similar to the attached flow module of the LB model. The second equation describes the reduced lift effect due to unsteady flow separation. Some modifications to the original model were suggested by Peters (Peters, 1985) and Petot (Petot, 1997). These improvements, however, are not included in the present





implementation. The ONERA model can be written as (Brouwer, 1990; Holierhoek et al., 2013; Khan, 2018):

$$C_{L_n}^D = C_{L_n}^{D1} + C_{L_n}^{D2} \tag{36}$$

$$C_{D_n}^D = C_{D_n}^{VISC} \tag{37}$$

$$C_{M_n}^D = C_{M_n}^{VISC} \tag{38}$$

with $C_L^{D1}$ and $C_L^{D2}$ being the first and second corrections, respectively, defined as:

$$\dot{C}_{L_n}^{D1} + \lambda_L C_{L_n}^{D1} = \lambda_L C_{L_n}^{INV} + (\lambda_L s_L + \sigma_L)\dot{\alpha}_n + s_L \ddot{\alpha}_n \tag{39}$$

and

$$\ddot{C}_{L_n}^{D2} + a_L \dot{C}_{L_n}^{D2} + r_L C_{L_n}^{D2} = -(r_L \Delta C_{L_n}^{INV} + e_L \Delta \dot{C}_{L_n}^{INV}) \tag{40}$$

where,

$$r_L = \left[ r_0 + r_2 (\Delta C_{L_n}^{INV})^2 \right]^2 \tag{41}$$

$$a_L = a_0 + a_2 (\Delta C_{L_n}^{INV})^2 \tag{42}$$

$$e_L = e_2 (\Delta C_{L_n}^{INV})^2 \tag{43}$$

The time derivative of the above equations are with respect to the non-dimensional time $s = 2Vt/c$. The constants of Equation (39) as listed in Section 2.7 ($\lambda_L$, $s_L$, $\sigma_L$) can be obtained from the flat plate database in the absence of unsteady airfoil data. In contrast, the constants of Equation (40) ($r_0$, $r_2$, $a_0$, $a_2$, $e_2$) should be adjusted based on the curve-fitting procedure to a measured data, making the model more empirical.

### 2.3 Snel 2nd order model

The history of the Snel's second order model (Snel, 1997) dates back to 1993 based on Truong's observation on dynamic lift coefficient characteristics (Truong, 1993). Truong proposed that the difference between the static and dynamic lift can be divided into two terms: the forcing frequency response and the higher frequency dynamics of a self-excited nature. The total dynamic response of the airfoil is formulated as:

$$C_{L_n}^D = C_{L_n}^{VISC} + \Delta C_{L_n}^{D1} + \Delta C_{L_n}^{D2} \tag{44}$$

$$C_{D_n}^D = C_{D_n}^{VISC} + \underset{\nearrow^0}{\Delta C_{D_n}^{D1}} + \underset{\nearrow^0}{\Delta C_{D_n}^{D2}} \tag{45}$$

$$C_{M_n}^D = C_{M_n}^{VISC} + \underset{\nearrow^0}{\Delta C_{M_n}^{D1}} + \underset{\nearrow^0}{\Delta C_{M_n}^{D2}} \tag{46}$$

with $D1$ and $D2$ being the first and second order corrections, respectively. The first correction is modeled using an ordinary differential equation (ODE) by applying a spring-damping like function as:

$$\tau \Delta \dot{C}_{L_n}^{D1} + K f_{10_n} \Delta C_{L_n}^{D1} = F_{1_n} \tag{47}$$



The frequency of the first-order corrected lift follows the frequency of the forcing term $F_1$. This term is based on the time derivative of the difference between the steady inviscid $C_{L_n}^{INV}$ and viscous lift coefficient $C_{L_n}^{VISC}$ of an airfoil ($\Delta C_{L_n}^{INV}$) as:

$$F_{1_n} = \tau \Delta \dot{C}_{L_n}^{INV} \qquad (48)$$

$$\Delta C_{L_n}^{INV} = C_{L_n}^{INV} - C_{L_n}^{VISC} = \frac{dC_L}{d\alpha}(\alpha_n - \alpha_0^{INV}) - C_{L_n}^{VISC} \qquad (49)$$

with $n$ and $dC_L/d\alpha$ are the current sample time and inviscid lift gradient, respectively. The time constant $\tau$ in the above
equation represents the time required for the flow to travel half a chord distance as:

$$\tau = \frac{c}{2V} \qquad (50)$$

The "stiffness" coefficient of the first order term $Kf_{10_n}$ can be expressed as:

$$Kf_{10_n} = \begin{cases} \dfrac{1 + 0.5\Delta C_{L_n}^{INV}}{8(1 + 60\tau\dot{\alpha}_n)}; & \text{if} \quad \dot{\alpha}_n C_{L_n}^{INV} \leq 0 \\[3mm] \dfrac{1 + 0.5\Delta C_{L_n}^{INV}}{8(1 + 80\tau\dot{\alpha}_n)}; & \text{if} \quad \dot{\alpha}_n C_{L_n}^{INV} > 0. \end{cases} \qquad (51)$$

As shown in (Faber, 2018), the above equation becomes numerically unstable if $\dot{\alpha}_n$ is large (increasing reduced frequency
above 0.1) for $\dot{\alpha}_n C_{L_n}^{INV} \leq 0$. The reason is that the denominator goes to zero and then negative, causing numerical integration instability. Thus, based on pure intuition the denominator value was set to a minimum of 2.0 in Ref. (Faber, 2018). In the present implementation, a similar approach is adopted but the limit differs. Instead, the minimum denominator value is limited to $1 \times 10^{-5}$, because it yields more physical results for several cases tested by the authors.

To incorporate the higher order frequency dynamics, a second order ODE is used to describe the second order correction
term. The general form may be written as:

$$\tau^2 \Delta \ddot{C}_{L_n}^{D2} + Kf_{21_n}\Delta \dot{C}_{L_n}^{D2} + Kf_{20_n}\Delta C_{L_n}^{D2} = F_{2_n} \qquad (52)$$

similar to the first order correction, the frequency of the higher order dynamics is determined by the forcing term $F_{2_n}$, defined as:

$$F_{2_n} = 0.1k_s(-0.15\Delta C_{L_n}^{INV} + 0.05\Delta \dot{C}_{L_n}^{INV}). \qquad (53)$$

It is noted that the value 0.1 as a constant was chosen according to Ref. (Adema et al., 2019). This is not a fixed value and can be adjusted based on the evaluated cases as seen in literature (Adema et al., 2019; Snel, 1997; Holierhoek et al., 2013; Faber, 2018; Khan, 2018). Variable $k_s$ represents the Strouhal number that is typically 0.2. The spring coefficient is given by

$$Kf_{20_n} = k_s^2[1 + 3(\Delta C_{L_n}^{D2})^2][1 + 3\dot{\alpha}_n^2] \qquad (54)$$

and the damping coefficient as

$$Kf_{21_n} = \begin{cases} 60\tau k_s[-0.01(\Delta C_{L_n}^{INV} - 0.5) + 2(\Delta C_{L_n}^{D2})^2]; & \text{if} \quad \dot{\alpha}_n > 0 \\[2mm] 2\tau k_s; & \text{if} \quad \dot{\alpha}_n \leq 0. \end{cases} \qquad (55)$$





## 2.4 Adema-Snel 2nd Order Model

The recently developed model of Adema (Adema et al., 2019) improves the original Snel model (Snel, 1997) in several aspects. Instead of using the lift coefficient ($C_L$), the normal force coefficient ($C_N$) is used, similar to the LB model (Leishman and Beddoes, 1989). The total dynamic response of the airfoil is formulated as:

$$C_{N_n}^{D} = C_{N_n}^{VISC} + \Delta C_{N_n}^{D1} + \Delta C_{N_n}^{D2} \tag{56}$$

$$C_{T_n}^{D} = C_{T_n}^{VISC} + \Delta C_{T_n}^{D1}{}^{0} + \Delta C_{T_n}^{D2}{}^{0} \tag{57}$$

$$C_{M_n}^{D} = C_{M_n}^{VISC} + \Delta C_{M_n}^{D1}{}^{0} + \Delta C_{M_n}^{D2}{}^{0} \tag{58}$$

The model introduces some modifications of the original model in terms of: (1) the projected Strouhal number, (2) the first order coefficient and (3) the second order coefficient. The mathematical formulation of the first order term of the model is listed as:

$$\tau \Delta \dot{C}_{N_n}^{D1} + K f_{10_n} \Delta C_{N_n}^{D1} = F_{1_n} \tag{59}$$

$$F_{1_n} = \tau \Delta \dot{C}_{N_n}^{INV} \tag{60}$$

$$\Delta C_{N_n}^{INV} = C_{N_n}^{INV} - C_{N_n}^{VISC} = \frac{dC_N}{d\alpha}(\alpha_n - \alpha_0^{INV}) - C_{N_n}^{VISC} \tag{61}$$

$$K f_{10_n} = \begin{cases} \dfrac{1 + 0.2\Delta C_{N_n}^{INV}}{8(1 + 60\tau \dot{\alpha}_n)}; & \text{if} \quad \dot{\alpha}_n C_{N_n}^{INV} \leq 0 \\[4mm] \dfrac{1 + 0.2\Delta C_{N_n}^{INV}}{8(1 + 80\tau \dot{\alpha}_n)}; & \text{if} \quad \dot{\alpha}_n C_{N_n}^{INV} > 0 \end{cases} \tag{62}$$

and for the second order correction term as

$$\tau^2 \Delta \ddot{C}_{N_n}^{D2} + K f_{21_n} \Delta \dot{C}_{N_n}^{D2} + K f_{20_n} \Delta C_{N_n}^{D2} = F_{2_n} \tag{63}$$

$$F_{2_n} = 0.01 k_s (-0.04\Delta C_{N_n}^{INV} + 1.5\tau \Delta \dot{C}_{N_n}^{INV}). \tag{64}$$

$$K f_{20_n} = 10(k_s \sin \alpha_n)^2 [1 + 3(\Delta C_{N_n}^{D2})^2][1 + 280^2 \tau^2 \dot{\alpha}_n^2] \tag{65}$$

$$K f_{21_n} = \begin{cases} 60\tau k_s[-0.01(\Delta C_{N_n}^{INV} - 0.5) + 2(\Delta C_{N_n}^{D2})^2]; & \text{if} \quad \dot{\alpha}_n > 0 \\[2mm] 60\tau k_s[-0.01(\Delta C_{N_n}^{INV} - 0.5) + 14(\Delta C_{N_n}^{D2})^2]; & \text{if} \quad \dot{\alpha}_n \leq 0 \end{cases} \tag{66}$$

One may notice that Equation (64) contains $\tau$ in the second term of the right hand side (RHS). This is intended to remove the dependency of the model to the velocity as the input parameter. The other main difference with the original model is also observed in Equation (65) where the Strouhal number is projected by $\sin \alpha_n$. At last, the downstroke motion of the second order term of Equation (66) is modified to enable vortex shedding effects.

To sum up the characteristics of above discussed state-of-the-art dynamic stall models, Table 1 lists the properties of each model and in which aspects the model can be improved further.





**Table 1.** Properties of the discussed state-of-the-art dynamic stall models.

| Model name | First/second order | Higher harmonics | Model $C_L$ | Model $C_D$ | Model $C_M$ |
|---|---|---|---|---|---|
| Leishman-Beddoes | first order | - | x | x | x |
| Onera | second order | - | x | - | - |
| Snel | second order | x | x | - | - |
| Adema-Snel | second order | x | x | x | - |

## 2.5   New 2nd order IAG model

The proposed IAG model is developed based on knowledge gained from four different models; Leishman-Beddoes, Snel, Adema-Snel and ONERA models with modifications. Similar to the modern models like those from Snel (and ONERA) and its derivatives, the present model is constructed by two main terms: the first order and second order corrections. The total
dynamic response of the airfoil is formulated as:

$$C_{L_n}^D = C_{L_n}^{D1} + \Delta C_{L_n}^{D2} \tag{67}$$

$$C_{D_n}^D = C_{D_n}^{D1} + \Delta C_{D_n}^{D2} \tag{68}$$

$$C_{M_n}^D = C_{M_n}^{D1} + \Delta C_{M_n}^{D2} \tag{69}$$

with $^{D1}$ and $^{D2}$ being the first and second order corrections, respectively. Below the description of the modifications done for
the new model will be discussed in detail.

### 2.5.1   First order correction

Based on the Hopf-Biffurcation model of Truong (Truong, 1993) that used the LB-Model as the starting point of the first order correction, the present model does similarly. Despite that, the LB model is not transferred into the state-space formulation, but it is retained as the indical formulation. The model applies the superposition of the solution using a finite-difference
approximation to Duhamel's integral to construct the cumulative effect to an arbitrary time history of angle of attack. The LB-model described in Sections 2.1.1 to 2.1.3 will be used with the following modifications:

In the above LB-Model, predictions for drag is not accurate as will be shown in Section 3.2. This inaccuracy lies in the determination of $\eta$ in Equation (19) for the tangential force component because drag is more sensitive to tangential force than the lift force does. Also to maintain simplicity, parameter $\eta$ is removed and the tangential force is obtained from the static data
at the time-lagged angle of attack $\alpha_{f_n}$ by:

$$C_{T_n}^f = C_T^{VISC}(\alpha_{f_n}). \tag{70}$$

If one uses this formulation directly, at some point drag still becomes lower than the static drag value by a significant amount. By evaluating the experimental data for several airfoils and various flow conditions, this is not physical below the static stall





angle especially in the downstroke regime, where it usually just returns to the static value. In fact, those experimental data infer

that strong drag hysteresis occurs only at high angles of attack beyond stall. Therefore, a simple limiting factor is applied by

$$C_{D_n}^D = \begin{cases} C_{D_n}^{VISC}; & \text{if} \quad \alpha_n < \alpha_n^{CRIT} \quad \text{and} \quad C_{D_n}^D < C_{D_n}^{VISC} \\ C_{D_n}^D; & \text{otherwise} \end{cases} \tag{71}$$

The effects of these modifications are displayed in Figure 3.

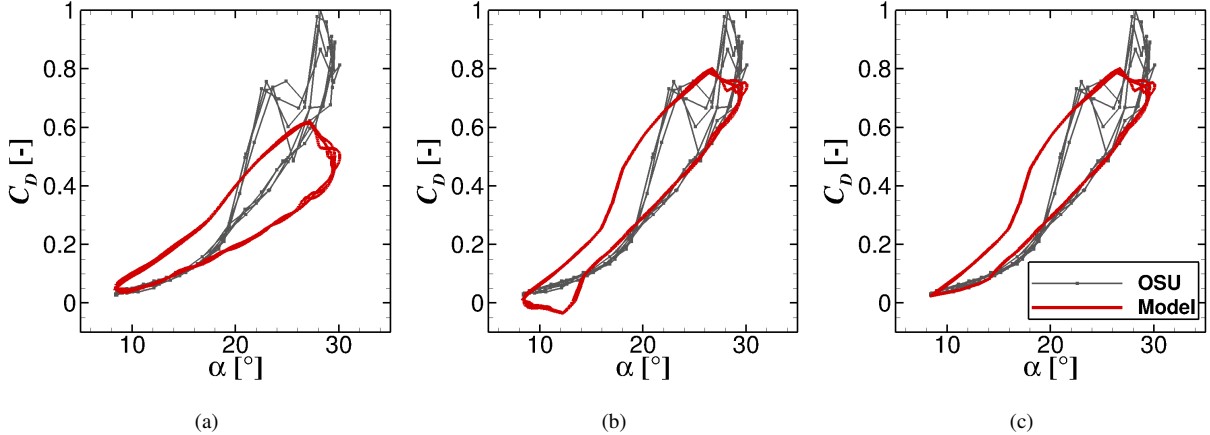

| (a) | (b) | (c) |

**Figure 3.** Drag reconstruction in comparison with the experimental data for S801 airfoil (Ramsay et al., 1996) applying: (a) Equation (19), (b) Equation (70) and (c) Equations (70) + (71).

It will also be shown in Section 3.2, that predicting moment coefficient directly from the static polar data by means of the time-lagged angle of attack has its drawback in the correct damping effect calculation. One may obtain better results by using

the "fitting function" as in Equation (20) instead of using Equation (34). However, this limits the usability for different airfoils, since the fitting has to be done again for each individual airfoil. For wind turbine simulations, this is fairly impractical because a wind turbine blade is usually constructed by several different airfoils, not to mention the interpolated shapes in between each airfoil position. To solve for this issue, a relatively simple approach is introduced by applying a time delay on the circulatory moment response as:

$$C_{M_n}^C = \begin{cases} C_{M_{n-1}}^C \exp\left(-\dfrac{\Delta s}{T_M^U}\right) - C_{Pf_n}\left(C_{V_n} - C_{V_{n-1}}\right)\exp\left(-\dfrac{\Delta s}{2T_M^U}\right); & \text{if} \quad \tau_{v_n} < T_{vl} \quad \text{and} \quad \Delta\alpha_n \geq 0 \\ C_{M_{n-1}}^C \exp\left(-\dfrac{\Delta s}{T_M^D}\right) - C_{Pf_n}\left(C_{V_n} - C_{V_{n-1}}\right)\exp\left(-\dfrac{\Delta s}{2T_M^D}\right); & \text{if} \quad \Delta\alpha_n < 0 \\ C_{M_{n-1}}^C; & \text{otherwise} \end{cases} \tag{72}$$

where,

$$C_{Pf_n} = K_f^C \frac{dC_N}{d\alpha}(\alpha_n^{CRIT} - \alpha_0^{INV}). \tag{73}$$





with $K_f^C$, $T_M^U$ and $T_M^D$ being constants relatively insensitive to airfoils. Furthermore, the second condition of Equation (35) is modified to avoid discontinuity which occurs at a large reduced frequency (eg. $k = 0.2$), for any LB-based models without re-calibration of the time constant as:

$$\tau_{v_n} = \begin{cases} \tau_{v_{n-1}} + 0.45 \dfrac{\Delta t}{c} V; & \text{if} \quad C_{N_n}^{P1} > C_N^{CRIT} \\ \tau_{v_{n-1}} \exp(-\Delta s); & \text{if} \quad C_{N_n}^{P1} < C_N^{CRIT} \quad \text{and} \quad \Delta \alpha_n \geq 0 \\ \tau_{v_{n-1}}; & \text{otherwise} \end{cases} \tag{74}$$

The effects of these modifications are displayed in Figure 4.

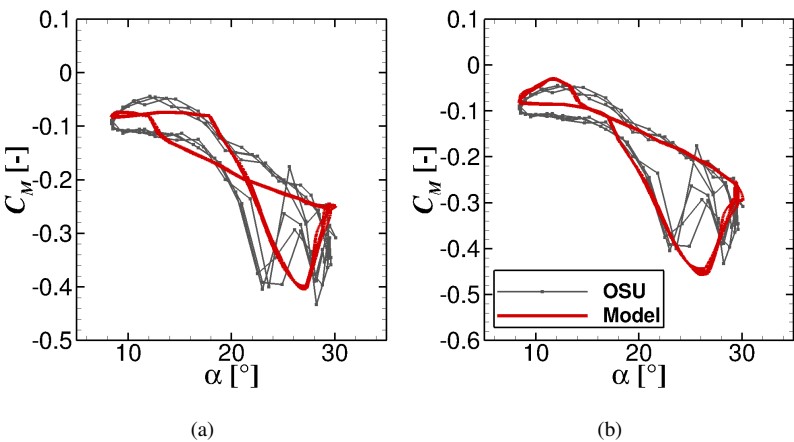

(a)                                        (b)

**Figure 4.** Moment reconstruction in comparison with the experimental data for S801 airfoil (Ramsay et al., 1996) applying: (a) Equation (34) and (b) Equation (72).

The total first order dynamic response of the airfoil is formulated as:

$$C_{N_n}^{D1} = C_{N_n}^f + C_{N_n}^V \tag{75}$$

$$C_{T_n}^{D1} = C_{T_n}^f \tag{76}$$

$$C_{M_n}^{D1} = C_{M_n}^f + C_{M_n}^V + C_{M_n}^C \tag{77}$$

where the definition and description of each variable was given in Section 2.1 for the LB model. Thus the first order lift and drag responses can be obtained by

$$C_{L_n}^{D1} = C_{N_n}^{D1} \cos \alpha_n - C_{T_n}^{D1} \sin \alpha_n \tag{78}$$

$$C_{D_n}^{D1} = C_{N_n}^{D1} \sin \alpha_n + C_{T_n}^{D1} \cos \alpha_n \tag{79}$$

### 2.5.2 Second order correction

The second order correction takes the form of the non-linear ordinary differential equation according to the second order correction of the Snel (Snel, 1997) and Adema-Snel (Adema et al., 2019) models. Generally, the basis of implementation of





the present model mostly uses the Adema-Snel (Adema et al., 2019) model where the normal force is used instead of the lift

force as for the original Snel model (Snel, 1997) as:

$$\Delta \ddot{C}_{N_n}^{D2} + Kf_{21_n}\Delta \dot{C}_{N_n}^{D2} + Kf_{20_n}\Delta C_{N_n}^{D2} = F_{2_n} \tag{80}$$

This way, shedding effects apply not only on the lift force but also on the drag force. Note that $\tau$ is not present in Equation (80) in contrast to the original formulation in Equations (52) and (63). The equation is changed because the time derivatives in the above equation is no longer with respect to time but to $s = 2Vt/c$, similar to the ONERA model (Tran and Petot, 1980; Dat

and Tran, 1981; Petot, 1989). This is done to nondimensionalize the equations.

In Equation (65), the Strouhal number $k_s$ was projected as a function of the angle of attack by $\sin\alpha_n$. This modification causes problem when the hysteresis effect takes place in both positive and negative angles because Equation (65) will be zero and then negative, causing instability of the ODE. Thus, the original form of the Snel model (Snel, 1997) is retained in the present model, but the constant is modified as.

$$Kf_{20_n} = 20k_s^2[1 + 3(\Delta C_{N_n}^{D2})^2][1 + 3\dot{\alpha_n}^2] \tag{81}$$

The idea for the downtroke damping as in Equation (66) is adopted in the present model, the following form and constants are used:

$$Kf_{21_n} = \begin{cases} 150k_s[-0.01(\Delta C_{N_n}^{INV} - 0.5) + 2(\Delta C_{N_n}^{D2})^2]; & \text{if} \quad \dot{\alpha_n} > 0 \\ 30k_s[-0.01(\Delta C_{N_n}^{INV} - 0.5) + 14(\Delta C_{N_n}^{D2})^2]; & \text{if} \quad \dot{\alpha_n} \le 0 \end{cases} \tag{82}$$

Note again that $\tau$ is not present in the above equation. As for the forcing term, the original form of the Snel model (Snel, 1997)

is adopted as:

$$F_{2_n} = 0.5k_s(-0.15\Delta C_{N_n}^{INV} + 0.05\dot{\Delta C}_{N_n}^{INV}). \tag{83}$$

To facilitate the inclusion of the higher harmonic effects for the pitching moment, the idealized center of pressure obtained in the first order correction given in Equation (26) is passed into the second order model. Thus, the dynamic moment reaction takes the form:

$$\Delta C_{M_n}^{D2} = -C_{Pv_n}\Delta C_{N_n}^{D2} \tag{84}$$

Regarding the tangential force, a similar assumption is made as in Equation (56) where the influence of $\Delta C_{T_n}^{D2}$ is neglected in the formulation. Finally, the second order term of the lift ($\Delta C_{L_n}^{D2}$) and drag ($\Delta C_{D_n}^{D2}$) can be calculated easily. The effects of the second order term are displayed in Figure 5.



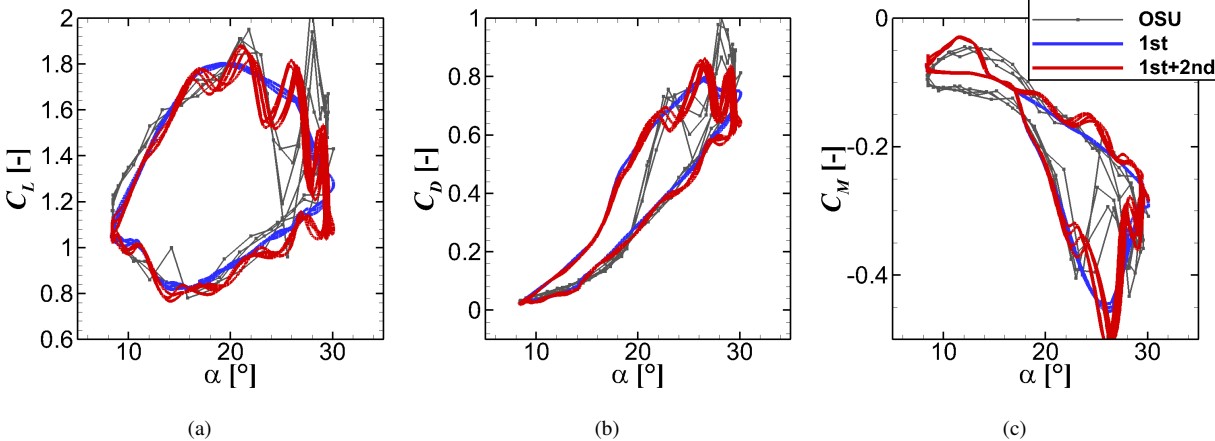

**Figure 5.** Airfoils response reconstruction in comparison with the experimental data for S801 airfoil (Ramsay et al., 1996) applying only the first order correction and with inclusion of the second order term. (a) Lift, (b) drag and (c) pitching moment.

## 2.6 Static polar reconstruction

Larsen et al. (Larsen et al., 2007) demonstrated that the inviscid lift ($C_{Li}$) formulation in the Leishman-Beddoes model (Leishman and Beddoes, 1989) can be problematic for the static lift reproduction at large angles of attack as depicted in Figure 6 (Larsen et al., 2007). The employed correction for the normal force reads:

$$C_{N_n}^{INV} = \begin{cases} \dfrac{dC_N}{d\alpha}(\alpha_n - \alpha_0^{INV})); & \text{if} \quad f_n > 0 \\ 4C_{N_n}^{VISC}; & \text{if} \quad f_n = 0 \end{cases} \tag{85}$$

in contrast to the original formulation that reads (Leishman and Beddoes, 1989):

$$C_{N_n}^{INV} = 2\pi(\alpha_n - \alpha_0^{INV}) \tag{86}$$

Note that $dC_N/\alpha$ of $2\pi$ was used in Ref. (Larsen et al., 2007). In the present work, the maximum inviscid polar gradient is used instead according to the recommendation of Hansen et al. (Hansen et al., 2004). This approach was implemented and tested for vertical axis wind turbine modeling in (Bangga et al., 2019). This relation is applied for all tested dynamic stall models in this report.



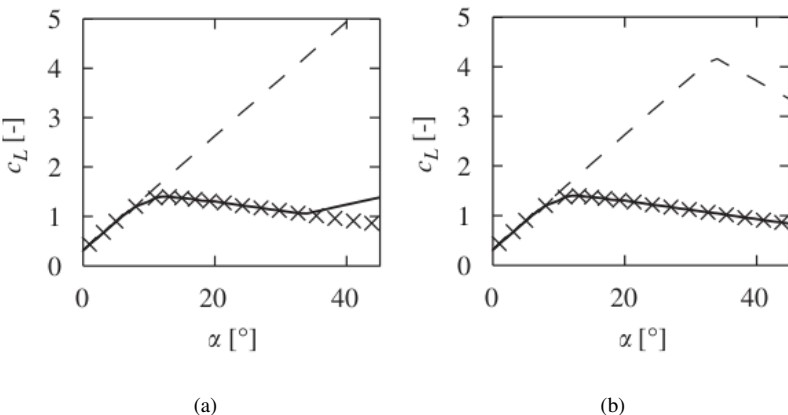

|  (a)  |  (b)  |

**Figure 6.** Modeled and measured static lift representation from a NACA 63-418 profile (Abbott and Von Doenhoff, 1959). (a) Leishman-Beddoes (Leishman and Beddoes, 1989) formulation based on the Kirchhoff flow theory and (b) corrected model from Larsen et al. (Larsen et al., 2007); – – inviscsid lift, — reproduced viscous lift, × experiment (Larsen et al., 2007).

## 2.7 Constants applied for the investigated dynamic stall models

The following constants are applied in the implemented dynamic stall models. These values are kept constant throughout the paper. The constants for the Leishman-Beddoes model are given in Table 2, for the ONERA model in Table 3 and for the proposed IAG model in Table 4. For any model developed based on the Leishman-Beddoes type, the critical angle of attack plays a major role. This can be obtained as the angle where the viscous pitching moment breaks or when the drag increases
significantly. The applied critical angles are given in Table 5.

**Table 2.** Constants applied for the Leishman-Beddoes model.

| $A_1$ | $A_2$ | $b_1$ | $b_2$ | $K_\alpha$ | $T_p$ | $T_f$ | $T_v$ | $T_{vl}$ | $K_v$ | $\eta$ |
|---|---|---|---|---|---|---|---|---|---|---|
| 0.3 | 0.7 | 0.14 | 0.53 | 0.75 | 1.7 | 3.0 | 6.0 | 6.0 | 0.2 | 0.95 |

**Table 3.** Constants applied for the ONERA model.

| $\lambda_L$ | $s_L$ | $\sigma_L$ | $r_0$ | $r_2$ | $a_0$ | $a_2$ | $e_2$ |
|---|---|---|---|---|---|---|---|
| 0.17 | $\pi$ | $2\pi$ | 0.2 | 0.2 | 0.3 | 0.2 | -0.35 |





**Table 4.** Constants applied for the IAG model.

| $A_1$ | $A_2$ | $b_1$ | $b_2$ | $K_\alpha$ | $T_p$ | $T_f$ | $T_v$ | $T_{vl}$ | $K_v$ | $K_f^C$ | $T_m^U$ | $T_m^D$ |
|---|---|---|---|---|---|---|---|---|---|---|---|---|
| 0.3 | 0.7 | 0.7 | 0.53 | 0.75 | 1.7 | 3.0 | 6.0 | 6.0 | 0.2 | 0.1 | 1.5 | 1.5 |

**Table 5.** Critical angle of attack ($\alpha_n^{CRIT}$) applied for the Leishman-Beddoes and IAG models.

| S801 | NACA4415 | S809 | S814 |
|---|---|---|---|
| 15.1° | 15.1° | 14.1° | 10° |

## 3 Results and Discussion

The four state-of-the-art dynamic stall models discussed above (Leishman-Beddoes, ONERA, Snel, Adema-Snel) have been used as a basis for examining the dynamic loads of four different pitching airfoils at various flow conditions. Experience gained from those models is used to formulate a new 2nd order dynamic stall model, namely IAG model by evaluating the weakness

and strength of each model. Except for the Leishman-Beddoes model, all presented models need to solve a set of differential equations. For this purpose, the Euler-Heun forward integration method is used. The validation is done by comparing the calculations with experimental data performed at the Ohio State University for the S801 airfoil (13.5% relative thickness) (Ramsay et al., 1996), NACA4415 airfoil (15% relative thickness) (Hoffman et al., 1996), S809 airfoil (21% relative thickness) (Ramsay et al., 1995) and S814 airfoil (24% relative thickness) (Janiszewska et al., 1996). All selected test cases are for the

airfoils employed with a leading edge grit (turbulator) to enable the "soiled" effects on a wind turbine blade at a Reynolds number of around 750K. The results of the studies are presented in the following sections.

### 3.1 Sensitivity against applied time step size

In this section, the sensitivity of the investigated models to changes in the time step size is evaluated. In practice, time step is an important aspect that determines the computational effort needed for BEM/vortex lattice method calculations in industry.

Usually, the a step of around $\Delta t \approx$ 0.003-0.05 s (20-300 Hz) is applied in practice. Note that a period of turbine rotation roughly corresponds to a period of pitching cycle of the airfoil section undergoing yawed or sheared inflow conditions. To better visualize the time step size within the OSU measurement, a sample of test case of the S801 airfoil at $k$ = 0.073 is considered. The measurement was conducted at a tunnel speed of 23.7 m/s and a chord length of 0.457 m, which can be transferred to a cycle period of $T = \pi c/(Vk)$ = 0.829 s or a frequency $f$ of 1.2 Hz. This information directly leads to a

conclusion that one cycle of dynamic stall is actually only resolved by a maximum of $300\Delta t$ in wind turbine simulations.

In Figure 7, the effects of various time step size on several dynamic stall models are shown. The changes obtained by varying the time step size stem from the integration of the differential equations, thus causing numerical uncertainty. It can be seen clearly that the Leishman-Beddoes, ONERA and Snel models are relatively insensitive to any changes in time step size,





which make the models suitable for calculations using coarser temporal discretization. However, those models lack of higher

harmonic fluctuations and might not perform well in the near/post stall conditions. On the other hand, the Adema model is able

to reproduce the higher harmonic effects, but the response is strongly dependent upon the time step size. In this regard, the IAG

model turns out to be less sensitive to temporal discretization variation.

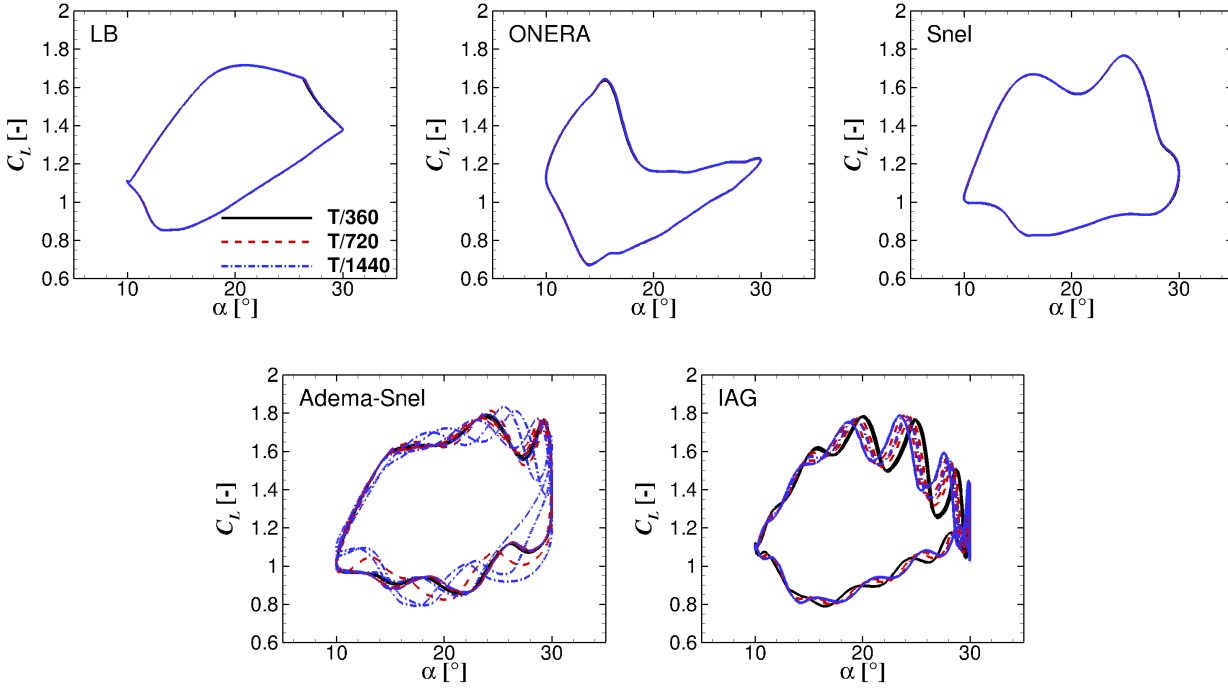

**Figure 7.** Sensitivity and stability of several dynamic stall models with time step size variation. $T$ represents the oscillation period. S801 airfoil, $k = 0.073$, $\overline{\alpha} = 20°$, $\Delta\alpha = 10°$.

## 3.2   Comparison against experimental data

While the effects of time step size are assessed in Section 3.1, this section compares the predicted dynamic forces in comparison

with the measurement data. For a fair comparison, all models are assessed with the same time step size of $\Delta t = T/1440$. The

evaluations are performed on the S801 airfoil at $k = 0.073$, at the same condition as already presented in Section 3.1. The

comparison of each model is shown in Figures 8 to 12 for the Leishman-Beddoes, ONERA, Snel, Adema and IAG models,

respectively. Note that the constants of the other four existing dynamic stall models are taken directly from literature without

further calibration for the S801 airfoil. Therefore, it is already expected that their performance will not be optimal. The main

purpose of the comparison is not to study their accuracy, but to analyze the robustness of each model for a different airfoil

without tuning the constants. On the other hand, the constants for the IAG model in Table 4 were obtained using the S801



airfoil. To enable a fair assessment on the model robustness, the IAG model will also be used to reconstruct the dynamic polar data of four airfoils with different relative thickness without changing the constants in Section 3.7.

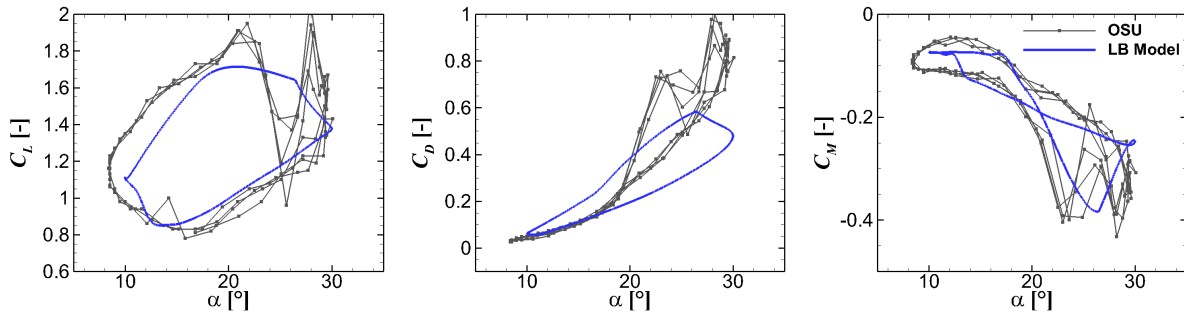

**Figure 8.** Dynamic forces reconstruction using the Leishman-Beddoes model in comparison with the measurement data (Ramsay et al., 1996) for $\Delta t = T/1440$. S801 airfoil, $k = 0.073$, $\overline{\alpha} = 20°$, $\Delta\alpha = 10°$.

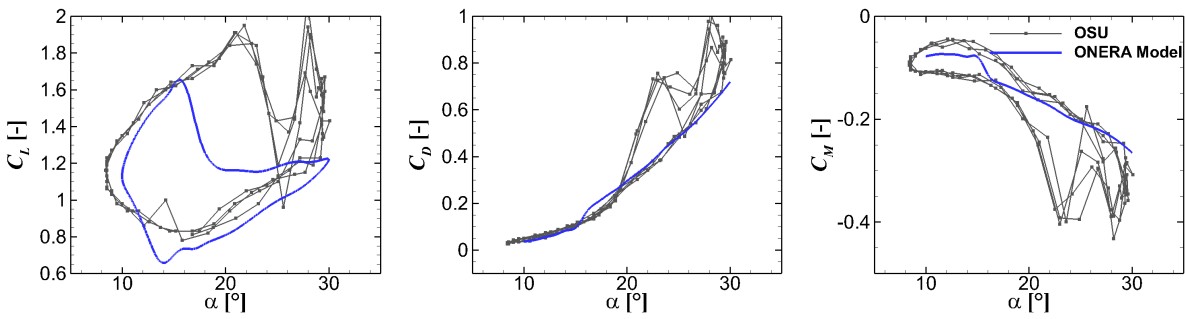

**Figure 9.** Dynamic forces reconstruction using the ONERA model in comparison with the measurement data (Ramsay et al., 1996) for $\Delta t = T/1440$. S801 airfoil, $k = 0.073$, $\overline{\alpha} = 20°$, $\Delta\alpha = 10°$.

As already well documented in several literature, the general trend of aerodynamic forces can be reproduced fairly well by
the Leishman-Beddoes model in Figure 8. However, the model cannot reproduce the higher order harmonics and shows some discrepancies on the predicted drag and moment coefficients, as it was documented in (Bangga et al., 2020). In contrast, the original ONERA and Snel models cannot predict the drag and moment coefficients in the original formulations. Thus, only the static polar data is shown. The ONERA model predicts earlier stall than the measured data. This might be caused by the flat plate assumption in its formulation. Determination of the constants also plays a significant role for the ONERA model.
As the constants in literature are applied in this paper without further tuning, the outcomes are already expected. On the other hand, this also indicates that the ONERA model is very sensitive to the applied constants, making it less robust for industrial purpose, where experimental/reference data is often not available. This is in contrast to the Snel model, which actually shows an acceptable accuracy even though the constants are taken as found in literature. The higher harmonic effects are unfortunately





not captured by this model. This is further refined by the Adema model which was developed as an improvement for the Snel
model. The model performs fairly well for the lift and drag predictions, though the drag value at small angles of attack is a bit

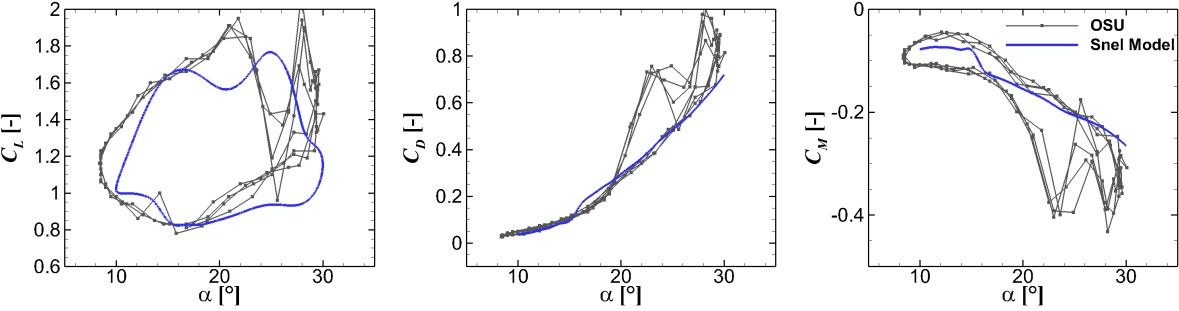

**Figure 10.** Dynamic forces reconstruction using the Snel model in comparison with the measurement data (Ramsay et al., 1996) for $\Delta t = T/1440$. S801 airfoil, $k = 0.073$, $\overline{\alpha} = 20°$, $\Delta\alpha = 10°$.

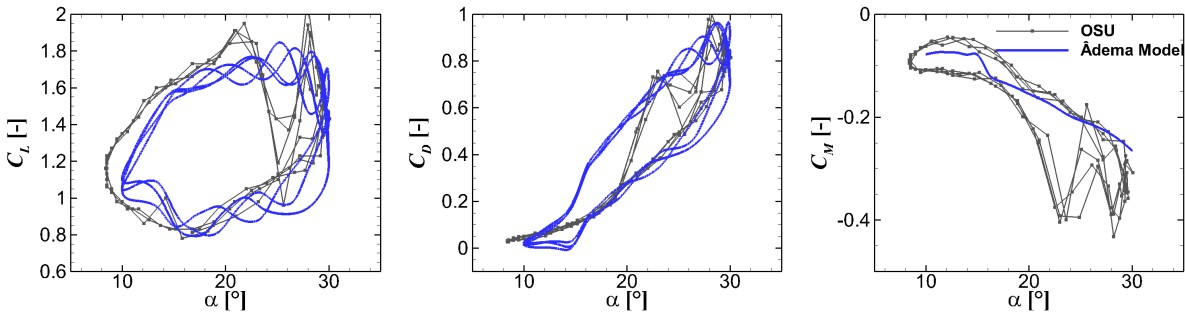

**Figure 11.** Dynamic forces reconstruction using the Adema model in comparison with the measurement data (Ramsay et al., 1996) for $\Delta t = T/1440$. S801 airfoil, $k = 0.073$, $\overline{\alpha} = 20°$, $\Delta\alpha = 10°$.

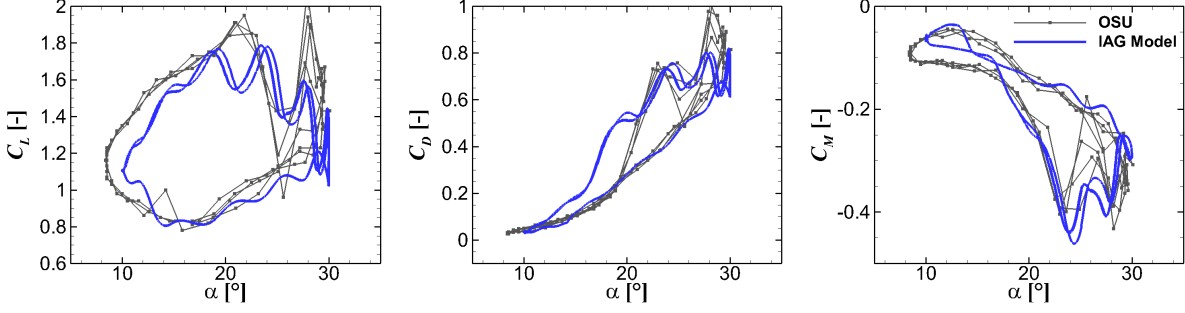

**Figure 12.** Dynamic forces reconstruction using the IAG model in comparison with the measurement data (Ramsay et al., 1996) for $\Delta t = T/1440$. S801 airfoil, $k = 0.073$, $\overline{\alpha} = 20°$, $\Delta\alpha = 10°$.



off. Again, the main drawback of the model is its dependency upon the time step size applied as already shown in Section 3.1. The pitching moment prediction is also not included in its formulations. These disadvantages are better treated in the proposed IAG model. Not only the prediction of the lift coefficient, but also the accuracy of drag prediction is improved significantly. The modifications described in Section 2.5 result in the improvement at low and high angles of attack regimes. The model is

also able to reconstruct the pitching moment polar accurately, which is important for aeroelastic calculations of wind turbine blades.

For the following sections, the proposed IAG model will be tested under various flow conditions and for several airfoils at various relative thicknesses in comparison with measurement data. Note that these calculations are performed without changing the constants to assess the robustness of the model at different flow conditions.

**3.3 Effects of time signal deviation**

The actual pitching motion within the OSU measurement differs slightly from the intended motion. The actual time series of the angle of attack is included in the experimental data (Ramsay et al., 1996; Hoffman et al., 1996; Ramsay et al., 1995; Janiszewska et al., 1996). To assess the effects of this time signal deviation on the aerodynamic response, the calculations using this time signal data were performed applying the IAG model. The results are compared with the experimental data and

the results of the IAG model presented in Section 3.2. Note that this time signal data is fairly coarse, and can cause problems for second order dynamic stall models because the gradient of $\alpha$ change can be extremely large. To cover for this issue, the time signal is interpolated in between each available point using a third-order cubic-spline interpolation. The time signals are discretized by $\Delta t = T/1440$ over a single pitching period. The first period of oscillation is used for initialization of the time integration, thus a constant angle of attack is applied as shown in Figure 13.

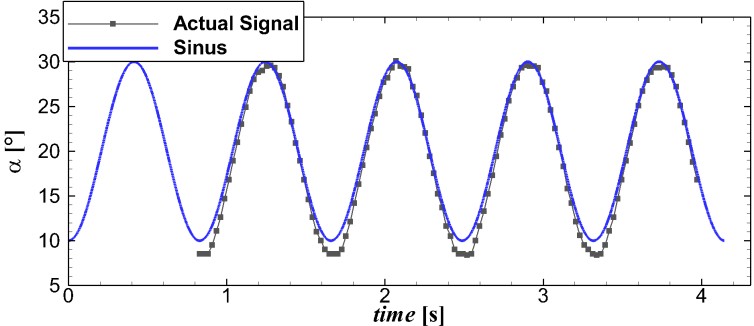

**Figure 13.** Comparison of the timeseries of the idealized sinusoidal angle of attack to the exact signals in the experimental campaign for S801 airfoil, $k = 0.073$, $\overline{\alpha} = 20°$, $\Delta\alpha = 10°$.

Figure 14 presents the influence of the time signal variation on the aerodynamic performance in terms of $C_L$, $C_D$ and $C_M$. TS labels the exact time signals in the experimental campaign. Although the time signal difference has almost no influence of the global prediction characteristics, some deviation with the idealized sinusoidal motion can be noticed clearly. For example, the increased lift build up in the upstroke regime before stall and the location of the lift stall. Some deviations on the drag and





pitching moment coefficients are observed as well. For the rest of the paper, the prediction procedure using the actual time

signal from the experimental data is used for best consistency with the experimental campaign.

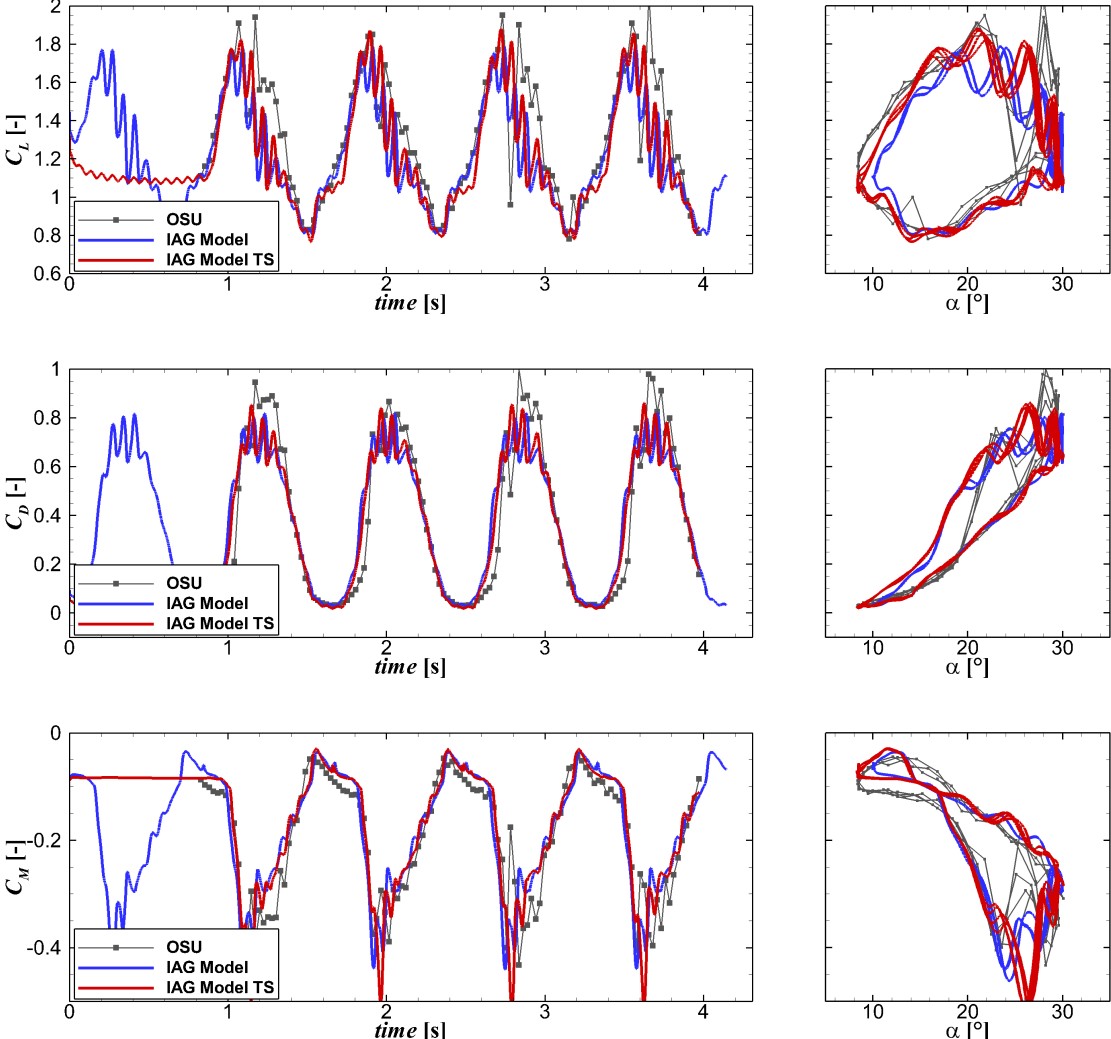

**Figure 14.** Dynamic forces reconstruction by the IAG model in comparison with the measurement data (Ramsay et al., 1996) for $\Delta t = T/1440$ using the actual angle of attack in the experimental campaign. TS labels the exact time signals in the experimental campaign. S801 airfoil, $k = 0.073$, $\overline{\alpha} = 20°$, $\Delta\alpha = 10°$.

### 3.4   Performance of the model for different mean angles of incidence

In this section, the effects of the mean angle of attack are evaluated. Three different angles of attack at the same inflow conditions are selected for this purpose. These are $\overline{\alpha} = 8°$, $14°$ and $20°$. Note that these mean angles of attack are only approximations




since the actual time signal data from the experimental campaign is used. The selected mean angles represent the regime of
attached flow, partly separated and fully separated flow conditions. These are helpful to assess the model performance under
various flow situations.

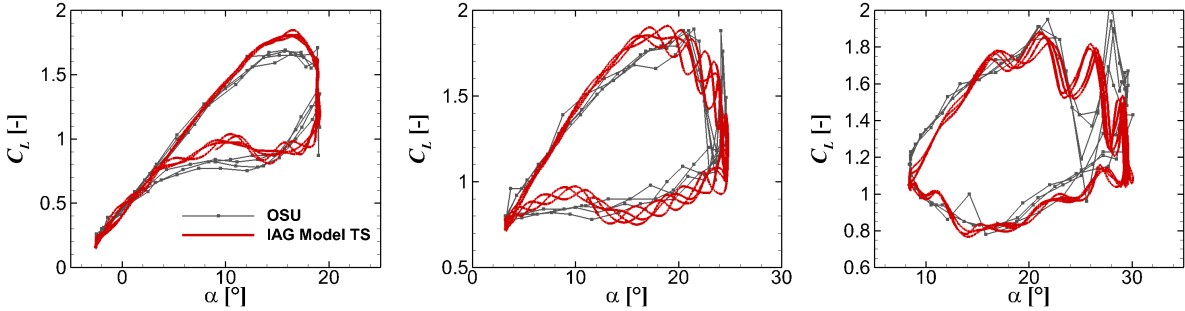

**Figure 15.** Lift reconstruction by the IAG model in comparison with the measurement data (Ramsay et al., 1996) for $\Delta t = T/1440$ using the actual angle of attack in the experimental campaign at various $\overline{\alpha}$. From left to right: $\overline{\alpha} = 8°$, $\overline{\alpha} = 14°$ and $\overline{\alpha} = 20°$. S801 airfoil, $k = 0.073$, $\Delta\alpha = 10°$.

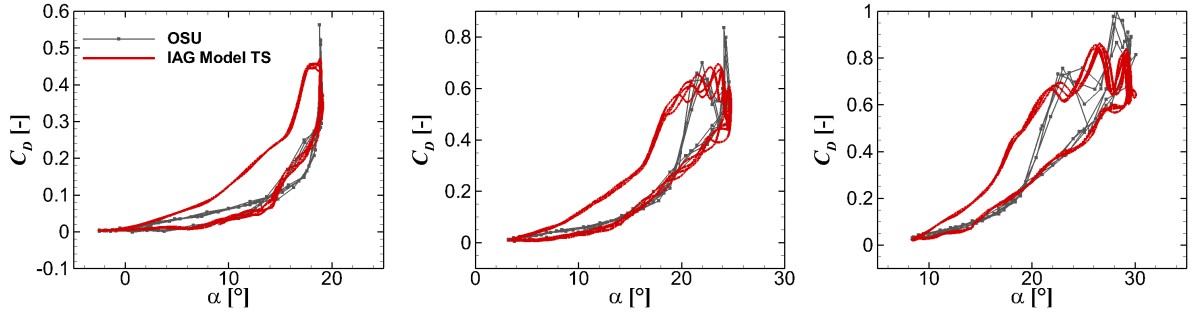

**Figure 16.** Drag reconstruction by the IAG model in comparison with the measurement data (Ramsay et al., 1996) for $\Delta t = T/1440$ using the actual angle of attack in the experimental campaign at various $\overline{\alpha}$. From left to right: $\overline{\alpha} = 8°$, $\overline{\alpha} = 14°$ and $\overline{\alpha} = 20°$. S801 airfoil, $k = 0.073$, $\Delta\alpha = 10°$

Figure 15 presents the results for the lift coefficient under these three investigated mean angles of attack. The model performs very well for these different cases. The maximum lift is a bit overestimated in the model for the lowest $\overline{\alpha}$, but in general all unsteady lift characteristics in the measurement data are reproduced in a sound agreement with the experimental data. A similar
behavior is shown for the drag prediction depicted in Figure 16. The proposed model captures the increased drag effect and its shedding characteristics well, though the drag coefficient is overestimated at small angles of attack in the upstroke phase (below the dynamic stall onset at $\alpha \approx 20°$). However, the simple modifications applied in Section 2.5 result in a good prediction of the downstroke drag coefficient as compared with the experimental data. In Figure 17, the prediction for pitching moment is shown. Here the predicted moment coefficient is in a good agreement with the measured values.



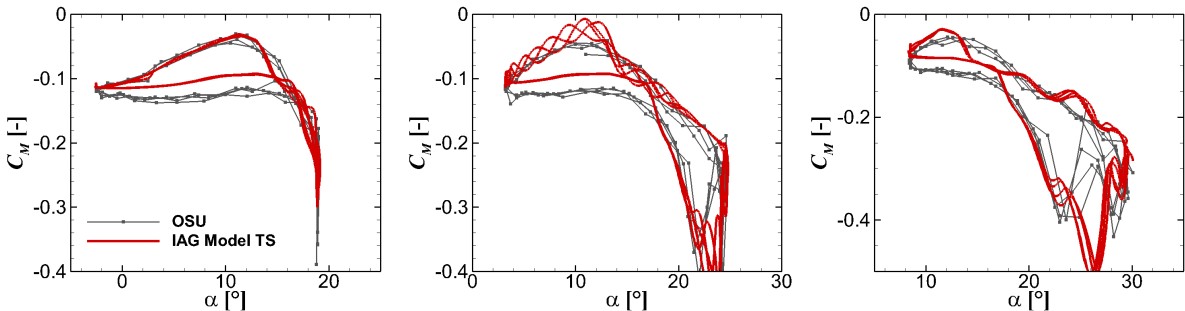

**Figure 17.** Pitching moment reconstruction by the IAG model in comparison with the measurement data (Ramsay et al., 1996) for $\Delta t = T/1440$ using the actual angle of attack in the experimental campaign at various $\overline{\alpha}$. From left to right: $\overline{\alpha} = 8°$, $\overline{\alpha} = 14°$ and $\overline{\alpha} = 20°$. S801 airfoil, $k = 0.073$, $\Delta\alpha = 10°$

## 3.5 Performance of the model for different reduced frequencies

The effects of pitching frequency on the aerodynamic response will be discussed in this section. Three different reduced frequencies are examined, namely $k = 0.036$, 0.073 and 0.111. The stall regime is shown here, where the prediction is the most challenging. The actual time signals as of the measurement campaign are used, following the procedure described in Section 3.3.

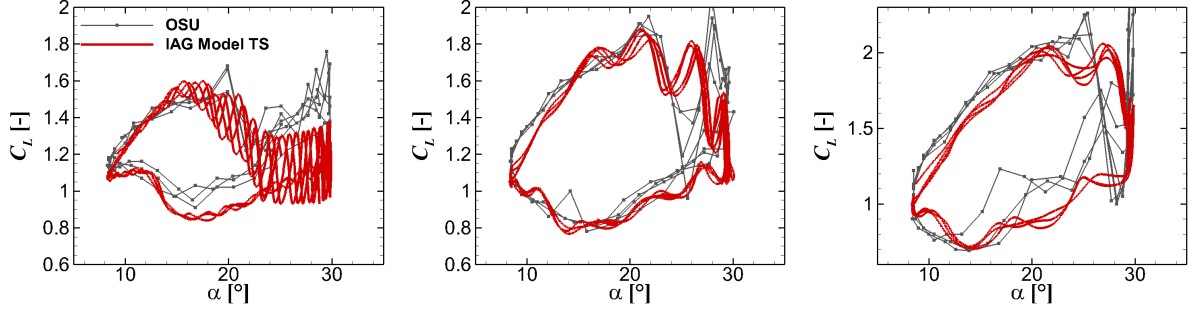

**Figure 18.** Lift reconstruction by the IAG model in comparison with the measurement data (Ramsay et al., 1996) for $\Delta t = T/1440$ using the actual angle of attack in the experimental campaign at various $k$. From left to right: $k = 0.036$, $k = 0.073$ and $k = 0.111$. S801 airfoil, $\overline{\alpha} = 20°$, $\Delta\alpha = 10°$.

Figure 18 displays the results for the dynamic lift coefficient response. The lowest reduced frequency of 0.036 is dominated by the viscous effects. It represents the case where the "delayed" angle of attack response is the weakest. It can be seen that the maximum attained lift coefficient increases with increasing $k$, which indicates the reduction of the "viscous" effects with increasing pitching frequency. The gradient of the lift polar in the upstroke and downstroke phase is also increasing as well.



These are followed by the reduction of the higher harmonics of the shedding frequency effects. These characteristics are present

in both experimental data and predictions delivered by the IAG model.

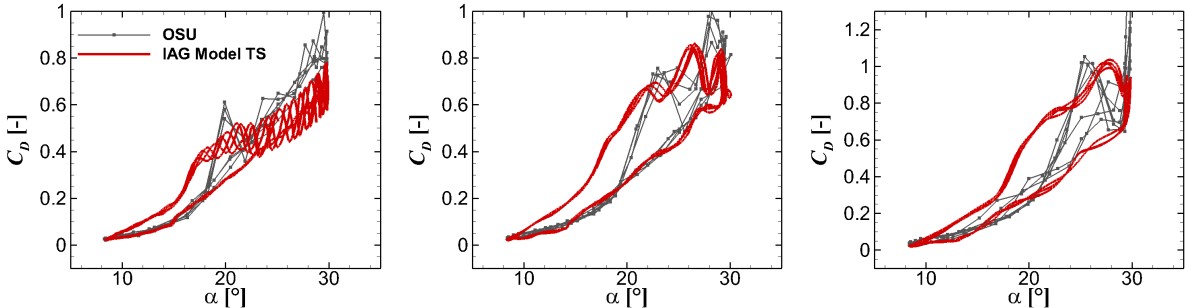

**Figure 19.** Drag reconstruction by the IAG model in comparison with the measurement data (Ramsay et al., 1996) for $\Delta t = T/1440$ using the actual angle of attack in the experimental campaign at various $k$. From left to right: $k = 0.036$, $k = 0.073$ and $k = 0.111$. S801 airfoil, $\overline{\alpha} = 20°$, $\Delta\alpha = 10°$.

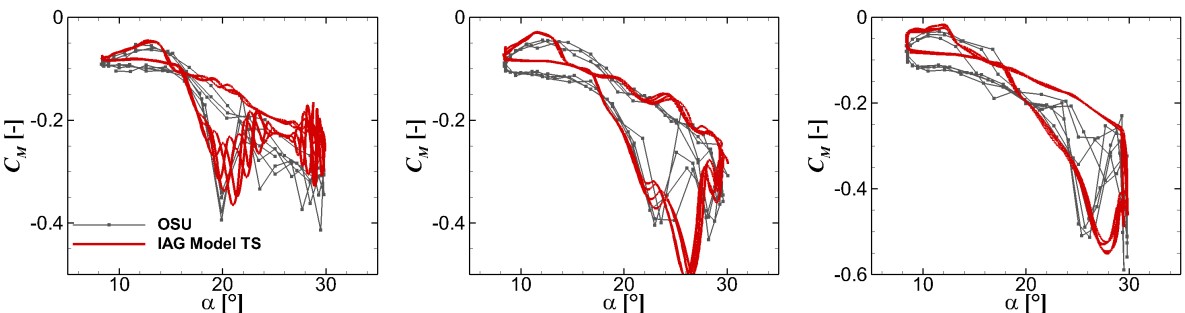

**Figure 20.** Pitching moment reconstruction by the IAG model in comparison with the measurement data (Ramsay et al., 1996) for $\Delta t = T/1440$ using the actual angle of attack in the experimental campaign at various $k$. From left to right: $k = 0.036$, $k = 0.073$ and $k = 0.111$. S801 airfoil, $\overline{\alpha} = 20°$, $\Delta\alpha = 10°$.

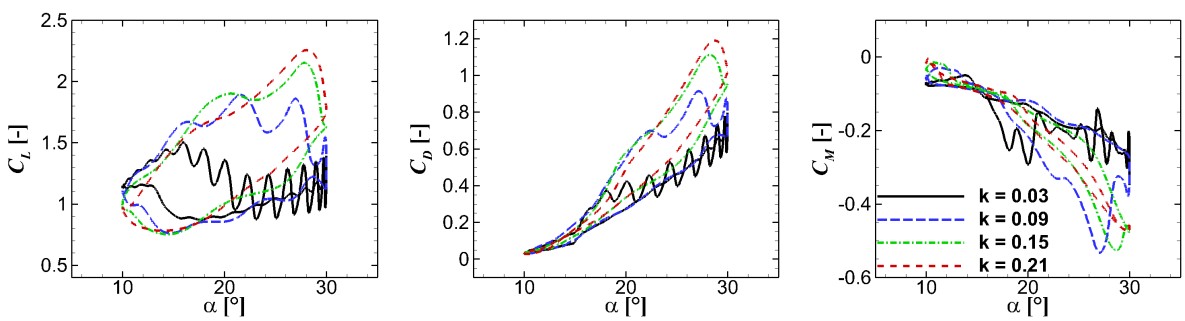

**Figure 21.** Effects of $k$ on the aerodynamic response by the IAG model for $\Delta t = T/1440$. S801 airfoil, $\overline{\alpha} = 20°$, $\Delta\alpha = 10°$.



A similar behavior is also displayed in drag and pitching moment in Figures 19 and 20, respectively. It is obvious that stall occurs much earlier for a smaller $k$ value. Interestingly, it is now observed that the hysteresis is smaller for $C_D$ and $C_M$ at $k = 0.111$ in comparison to the results at $k = 0.073$, both in experimental data and predictions. This effect is not clearly observed for $C_L$. The reason might be related to the reduced viscous effect at this large reduced frequency. This indicates that the polar

becomes more "inviscid" when $k$ increases above a certain value. To better investigate the limit of this hypothesis, the IAG model is used to reconstruct the dynamic polar data at various $k$ by applying an idealized sinusoidal motion as presented in Figure 21. Only the last DS cycle is shown for clarity of the observation. It can be seen clearly, that above $k = 0.1$ shedding effects are much weaker. Lift, drag and their gradient increase considerably. For the moment coefficient, the difference between the upstroke and downstroke value becomes much negligible with increasing $k$.

**3.6 Performance of the model for different pitching amplitudes**

In this section, the effects of pitching amplitude on the aerodynamic response of a pitching airfoil is investigated. The mean angle of attack is fixed at $\overline{\alpha} = 20°$. Note again that $\overline{\alpha}$ is only an approximation because the actual time signal data from the measurement campaign is applied. This large mean angle of attack is purposely selected because the post-stall characteristic is of interest and is well known for its violent vibration, even for the static condition. The small amplitude in this case means that

the whole pitch oscillation occurs within the stall regime.

Figures 22 to 24 display the dynamic force responses due to pitching motion of the airfoil predicted by the IAG model in comparison with the experimental data. The model accurately reconstructs the dynamic forces despite the predicted case is challenging within the post stall regime. Interesting to note is that the small pitching amplitude case induces stronger shedding effects than the larger amplitude case. This can be explained as following. As described by Leishman in his papers (Beddoes,

1982; Leishman, 1988; Leishman and Beddoes, 1989), the airfoil sees a lagged force response compared to the imposed disturbance. Therefore, in his model, a time-lagged angle of attack is introduced as the "effective" angle actually seen by the airfoil section. When the pitching motion takes place partly within the fully separated region (in the static case) and partly in

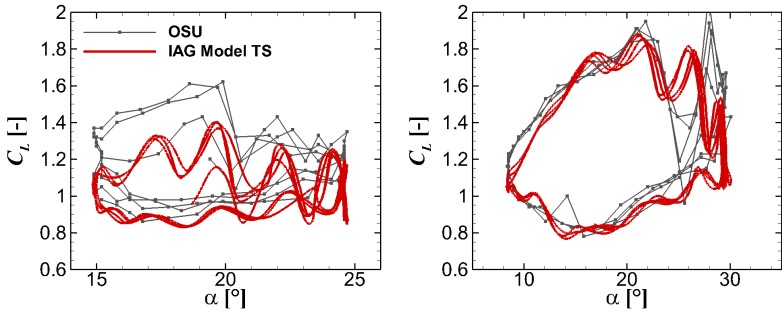

**Figure 22.** Lift reconstruction by the IAG model in comparison with the measurement data (Ramsay et al., 1996) for $\Delta t = T/1440$ using the actual angle of attack in the experimental campaign at various $\Delta\alpha$. Left: $\Delta\alpha = 5.5°$; right: $\Delta\alpha = 10°$. S801 airfoil, $k = 0.073$, $\overline{\alpha} = 20°$.



the attached/partly separated flow region, the airfoil still sees the lower angle (where the flow is still attached) even though
the pitching motion already reaches the post-stall regime. This effect stops/reduces when the effective angle is larger than the
critical angle defined in Table 5. As the critical angle for the S801 airfoil is defined at 15.1°, the lower amplitude case is fully
operating within the stall regime, where the attached flow effect is not present.

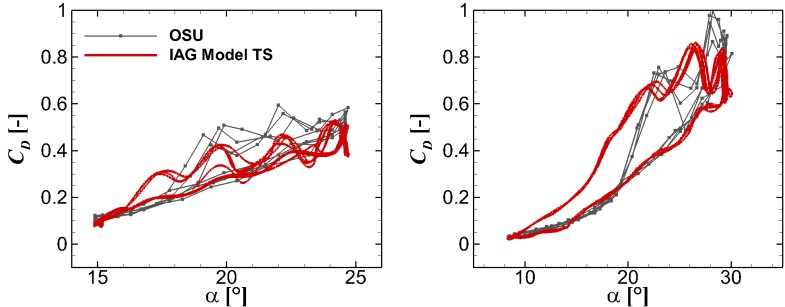

**Figure 23.** Drag reconstruction by the IAG model in comparison with the measurement data (Ramsay et al., 1996) for $\Delta t = T/1440$ using
the actual angle of attack in the experimental campaign at various $\Delta\alpha$. Left: $\Delta\alpha = 5.5°$; right: $\Delta\alpha = 10°$. S801 airfoil, $k = 0.073$, $\overline{\alpha} = 20°$.

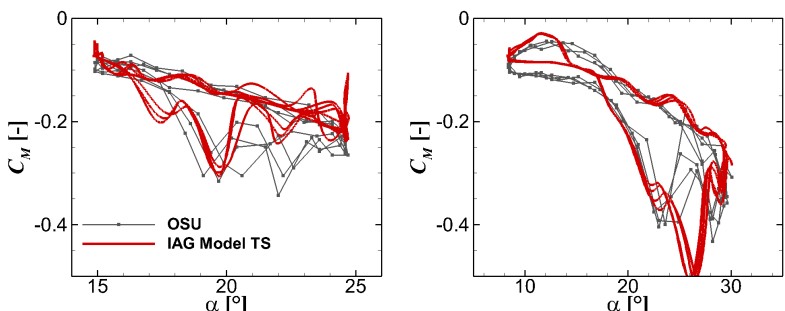

**Figure 24.** Pitching moment reconstruction by the IAG model in comparison with the measurement data (Ramsay et al., 1996) for $\Delta t =$
$T/1440$ using the actual angle of attack in the experimental campaign at various $\Delta\alpha$. Left: $\Delta\alpha = 5.5°$; right: $\Delta\alpha = 10°$. S801 airfoil,
$k = 0.073$, $\overline{\alpha} = 20°$.

### 3.7 Performance of the model for different airfoils

In this section, the performance and robustness of the proposed IAG model are assessed for airfoils with different relative
thickness. All model constants in Table 4 remain the same for all calculations. The difference from one airfoil calculation to
the other lies only in the critical angle of attack value as shown in Table 5. The value was obtained simply by looking at the
static polar data where the viscous pitching moment breaks or when the drag increases significantly.




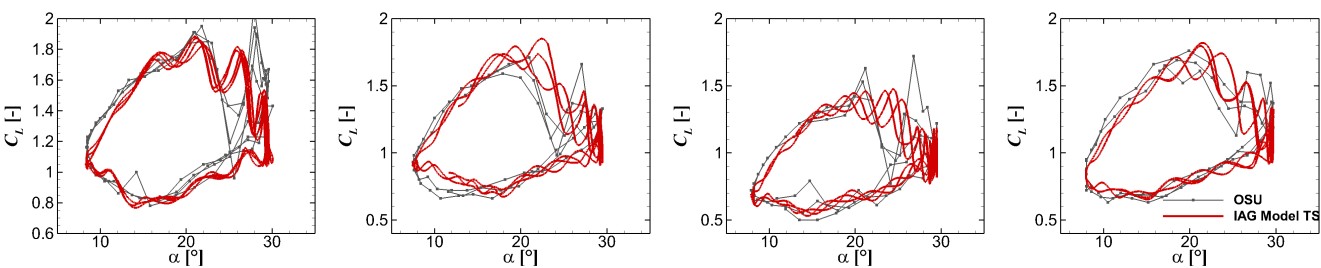

**Figure 25.** Lift reconstruction by the IAG model in comparison with the measurement data (Ramsay et al., 1996; Hoffman et al., 1996; Ramsay et al., 1995; Janiszewska et al., 1996) for $\Delta t = T/1440$ using the actual angle of attack in the experimental campaign for different airfoils. From left to right: S801 (13.5%), NACA4415 (15%), S809 (21%) and S814 (24%). $k = 0.073$, $\overline{\alpha} = 20°$, $\Delta\alpha = 10°$.

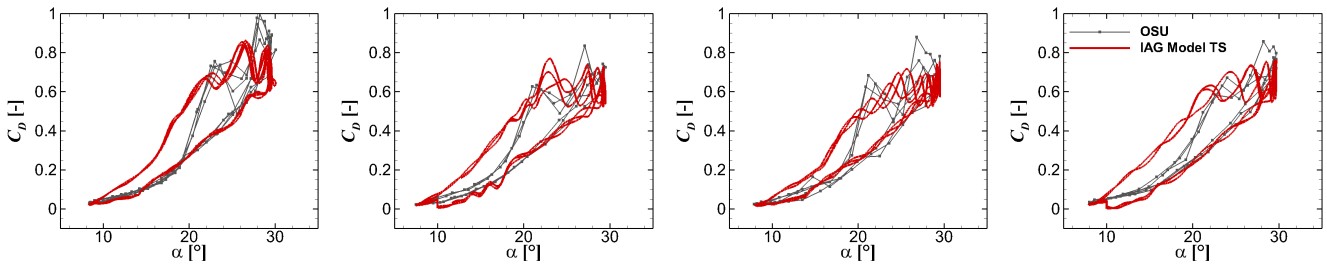

**Figure 26.** Drag reconstruction by the IAG model in comparison with the measurement data (Ramsay et al., 1996; Hoffman et al., 1996; Ramsay et al., 1995; Janiszewska et al., 1996) for $\Delta t = T/1440$ using the actual angle of attack in the experimental campaign for different airfoils. From left to right: S801 (13.5%), NACA4415 (15%), S809 (21%) and S814 (24%). $k = 0.073$, $\overline{\alpha} = 20°$, $\Delta\alpha = 10°$.

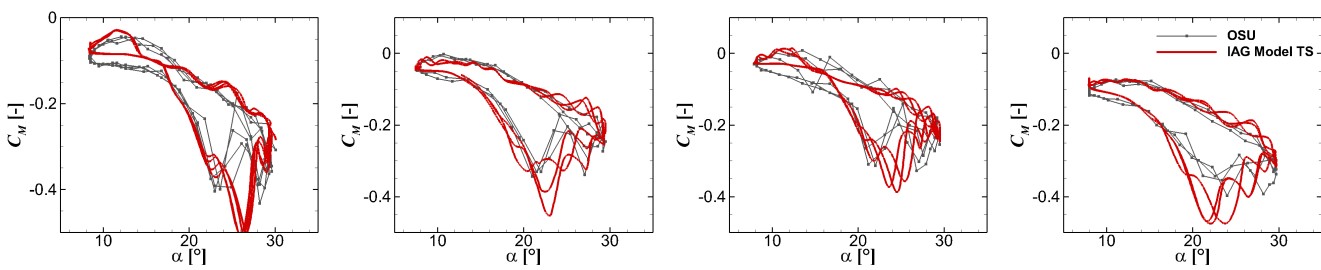

**Figure 27.** Pitching moment reconstruction by the IAG model in comparison with the measurement data (Ramsay et al., 1996; Hoffman et al., 1996; Ramsay et al., 1995; Janiszewska et al., 1996) for $\Delta t = T/1440$ using the actual angle of attack in the experimental campaign for different airfoils. From left to right: S801 (13.5%), NACA4415 (15%), S809 (21%) and S814 (24%). $k = 0.073$, $\overline{\alpha} = 20°$, $\Delta\alpha = 10°$.

**Figure 28.** Center of pressure reconstruction in comparison with the measurement data by the IAG model for $\Delta t = T/1440$ using the actual angle of attack in the experimental campaign for different airfoils. From top to bottom: S801 (13.5%), NACA4415 (15%), S809 (21%) and S814 (24%). $k = 0.073$, $\overline{\alpha} = 20°$, $\Delta\alpha = 10°$.



Despite the increased airfoil thickness from 13.5% to 24%, Figures 25 to 27 demonstrate that the reconstructed dynamic forces are in a good agreement with the experimental data, not only for the general trend but also the higher harmonic effects. As also the case for the Leishman-Beddoes model, it is important to select the appropriate value for the critical angle of attack.

The simple approach used in the present paper has shown its usefulness and potentially reduce the complexity of parameter tuning for industrial applications. Elgammi et al. (Elgammi and Sant, 2016) for example defined two different critical angles of attack, one for $C_N$ and the other for $C_T$ that were shown to improve the prediction accuracy. Although their attempt might be beneficial, this is not followed in this work because one main aim of the studies is to reduce parameter tuning required for one to the other cases.

### 3.8 Predictions of the center of pressure

To further complement the analyses conducted in Section 3.7, the location of the actual pressure center is calculated in this section as:

$$X_p = -\frac{C_M}{C_L} \tag{87}$$

which indicates the distance of the pressure point to the quarter chord position where $C_M$ is defined.

A correct location of the pressure point is important for determining the stability on aeroelastic simulations of wind turbine blades. The results of the calculations both for the experimental data and for the proposed IAG model are presented in Figures 28 for all four investigated airfoils both as time series and as the polar plot. It can be seen clearly that the agreement between the experimental data and the present predictions are excellent for all investigated airfoils.

## 4 Conclusions

Comprehensive studies on the accuracy of several state-of-the-art dynamic models to predict the aerodynamic loads of a pitching airfoil have been conducted. From the studies, the strength and weaknesses of each model were highlighted. This information was then transferred to develop a new second order dynamic stall model proposed in this paper. The new model improves the prediction for the aerodynamic forces and their higher harmonic effects due to vortex shedding, developed for robustness to improve its usability in practical wind turbine calculations. Details on the model characteristics, modifications

and treatment for numerical implementation were summarized in the present paper. The studies were conducted by examining the influence of the time step size, time signal deviation, mean angle of attack, reduced frequency, pitching amplitude and variation of the airfoil thickness. Several main conclusions can be drawn from the work.

– Time step size applied in wind turbine simulations is usually very coarse, the studies reveal that this has an influence on the consistency of the predicted loads especially for the second order dynamic stall models. The Leishman-Beddoes,

ONERA and Snel models are relatively insensitive to time step size variation. The Adema-Snel model shows the strongest time step dependency. In this regard, the IAG model is less sensitive to temporal discretization.



– The general characteristics of the polar data can be predicted by all investigated dynamic stall models. Despite that, only the Adema model and the present IAG model are able to demonstrate the higher harmonic effects.

– The exact time signal imposed based on the measurement campaign improves the prediction accuracy of the IAG model in comparsion with the idealized sinusoidal motion.

– The dynamic forces reconstructed by the IAG model are in a sound agreement with the experimental data under various flow conditions by variation of $\overline{\alpha}$, $k$, $\Delta\alpha$ and for four different airfoils without changing the constants.

– Increasing $k$ above 0.1 leads to an increased flow stability that reduces the viscous effects and vortex shedding influence.

– When the airfoil operates at a high $\overline{\alpha}$ within the stall regime, a small $\Delta\alpha$ leads to increased vibrations, because the time-lagged force response is weaker than for the larger $\Delta\alpha$ value.

*Author contributions.* G. Bangga developed the new model, designed the studies and conducted the analyses. T. Lutz and M. Arnold supported the research, provided suggestions and discussion about the manuscript.

*Competing interests.* The authors declare that they have no conflict of interest.

*Acknowledgements.* The authors gratefully acknowledge the Wobben Research and Development GmbH for providing the research funding
through the collaborative joint work DSWind. The measurement data provided from the Ohio State University is highly appreciated.





| **Variables** | |
|---|---|
| $s$ | nondimensional time |
| $V$ | incoming wind speed |
| $t$ | time |
| $c$ | chord |
| $f$ | separation factor |
| $C_N$ | normal force coefficient |
| $C_T$ | tangential force coefficient |
| $C_L$ | lift force coefficient |
| $C_D$ | drag force coefficient |
| $C_M$ | pitching moment coefficient |
| $C_N^P$ | total inviscid normal force coefficient |
| $C_N^P 1$ | time lagged total inviscid normal force coefficient |
| $C_N^I$ | impulsive inviscid normal force coefficient |
| $C_N^C$ | circulatory inviscid normal force coefficient |
| $C_N^f$ | viscous normal force coefficient |
| $C_T^f$ | viscous tangential force coefficient |
| $C_M^f$ | viscous pitching moment coefficient |
| $C_M^C$ | circulatory pitching moment coefficient |
| $C_N^V$ | vortex lift normal force coefficient |
| $C_N^{CRIT}$ | critical normal force coefficient |
| $X, Y, D_n$ | deficiency functions |
| $M$ | Mach number |
| $K_f$ | stiffness coefficient |
| $F_1$ | first order forcing term |
| $F_2$ | second order forcing term |

| **Greek letters** | |
|---|---|
| $\alpha$ | angle of attack |
| $\alpha_e$ | effective angle of attack |
| $\alpha_f$ | time lagged effective angle of attack |
| $\alpha^{CRIT}$ | critical angle of attack |
| $\beta$ | Mach number dependent parameter |
| $\tau_v$ | nondimensional vortex time |
| $\tau$ | time constant |



**Superscripts**

| | |
|---|---|
| $INV$ | static inviscid |
| $VISC$ | static viscous |
| $I$ | impulsive |
| $CRIT$ | critical |
| $D$ | dynamic loading |
| $D1$ | first order correction |
| $D2$ | second order correction |

**Subscripts**

| | |
|---|---|
| $n$ | present sampling time |
| $f$ | viscous lagged value |
| $v$ | vortex lift affected value |



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
