# Peer review of "An improved second order dynamic stall model for wind turbine airfoils"

_Wind Energy Science, 2020_

## Referee Comment (RC1) · Khiem V. Truong (Referee) · 19 May 2020

**1. General comments:**

The journal Wind Energy Science (WES) has recently accepted the publication of the paper "Development of a second order dynamic stall model" by N. Adema, M. Kloostermann and G. Schepers. This paper is about the improvement of Snel's model on vortex-shedding phenomena. This work answers to the current concern of industry for the design of Horizontal-axis wind turbines, due to the vibratory behavior of rotor blades in parked or idling conditions. The submitted paper for the same journal by Bangga et al. has for subject the same topic, the improvement of the second order equation of Snel's model. As it is submitted after the paper of Adema et al., it is confronted to

the challenge of providing significant results. Indeed, it embraces topics not treated in the recent paper quoted above, analysis of all the three aerodynamic coefficients (lift, drag and pitching moment) for four different airfoils in various flow conditions. The manuscript is well organized and clearly written. However, as the authors take on a new study field, they have not made a thorough analysis of the existing literature, and this fact leads to multiple errors in the submitted paper. Despite of these shortcomings, I still recommend its publication but with a major revision that takes into account the following critics.

2. Technical comments:

2.1. Analysis of the various stall models:

The authors spent a great length of time in analysis of the existing stall models that does not present a great interest for the manuscript objective. In doing so, the authors have made various mistakes. The Beddoes-Leishman model is not presented under the state-space formulation. Therefore, the sensitivity study of this model against step size of integration cannot be made, as stated in line 393. About the ONERA model, they are not aware that it was renamed ONERA-EDLIN ("Equations Différentielles Linéaires", meaning in English Linear Differential Equations), to distinguish it for the newer model ONERA-BH ("Bifurcation de Hopf", renamed later by his author as ONERA Hopf Bifurcation model). It is usual for researchers in the field of wind turbines to continue to call it with such name; so, this mistake is not serious. The critical error of the authors is to not consider the stall delay in the ONERA-EDLIN model. Without the account of stall delay, this model leads to predictions of the lift coefficient with large discrepancies in correlation with experiments as shown in Figure 9 p.20.

2.2. Values of constants used in the IAG model:

There are two types of constants used for the IAG model that are ill chosen, the critical stall angle and the value of the Strouhal number.

*The critical stall angle $\alpha$CRIT of airfoils is one key parameter for the stall model. The authors choose this value based on the position of the break of the pitching moment coefficient and the position of the important increase of the drag coefficient. This is not a good choice, as pointed by Sheng et al. in their conclusions (Reference cited on line 598), the best choice is the incidence angle at the maximum chord force coefficient. Led by such bad criteria of defining $\alpha$CRIT , the authors found very small values for the airfoils S801, S809 and S814 : 15.1o, 14.1o and 10o respectively, instead of the values of 17.6o , 19.2o and 13.9o found by Sheng et al. (Reference Sheng W., Galbraith R.A.McD. and Coton F.N., "Applications of low-speed dynamic-stall model to the NREL airfoils", Journal of Solar Energy Engineering, 2010, vol. 132, pp. 011006-1:011006-8). The increase of the value of $\alpha$CRIT would allow a better correlation of their model predictions with experiments, as shown in Figures 10-12, and following.

* Value of the Strouhal number S: the authors following Adema et al. use the value of S = 0.2; they should notice from various references that S is in the range of [0.06,0.13] (see for example "Spectral analysis of New MEXICO standstill measurements to investigate vortex shedding in deep stall" by Khan M.A., Ferreira C.S., Schepers G.J. and SØrensen N.N., Wind Energy, 2019, pp.1-14). When S decreases, the predicted distance between two consecutive extremum (maximum for the lift and drag coefficients, minimum for the pitching moment coefficient) of the aerodynamic coefficients increases. The correlation between model predictions and experiments would be improved.

2.3. Sensitivity of the results against applied time step of the solver:

The authors use a rudimentary numerical tool for solving the ordinary differential equations (ODE) with fixed time step, there exist more robust ODE solver with automatic step variation. Therefore, the discussion related to the time step size is irrelevant (section 3.1 and conclusion).

2.4. Quality of the IAG model:

The authors claim the superiority of their model over the others, but their model errors are not quantified. Since the study of Holierhoek et al. (cited in line 574), practically all the publications on hysteresis loops in stalled conditions of airfoils provide the values of the error L2-norm, see for instance the publication of Adema et al. I would consider that the predictions for the lift coefficient are reasonable. However, the predictions of the drag coefficient are overestimated and this would lead to under-prediction of the power coefficient CQ. The predictions of the pitching moment coefficient are not right in some cases. For instance for the airfoil S801 in Figure 17, the predictions show clockwise hysteresis sub-loop that correspond to negative aerodynamic damping, while the experiments show anti-clockwise sub-loop leading to positive aerodynamic damping.

2.5. The study of various airfoils:

It would be interesting that the model predictions could show some distinctive features associated with the thickness for the airfoils studied, ranging from thin (S801) to thick airfoil (S814). Thin airfoils are characterized by leading-edge stall, whereas thick airfoils by trailing-edge stall. The choice of the airfoil S801 by the authors of the submitted paper for extensive studies is unfortunate, because it is a thin airfoil of thickness 13.5%, and wind turbine blades have usually thickness larger than 15%.

3. Technical corrections:

Line 4: the sentence "many flow parameters" is not clear.

Line 538: "Increasing k above 0.1 leads to an increased flow stability": this is incorrect.

Lines 539-540: the assertion is incorrect. For a large angle variation, the variation of the pitching moment coefficient is more important (see Figure 24) and this could lead to more structural damage to the blades.

Imprecision for the section References: - Lines 555, 558, 566, 570, 577, 585, 598 and 604. - Inconsistencies for Lines 568 and 574

4. Concluding remarks and suggestions for revision:

Though the submitted paper is marred with errors, there are two positive aspects. The first one is about the objective of examining Snel's model for various flow conditions and airfoils. This stall model has been around 1997 and no exhaustive evaluation has been made at my knowledge. I feel that Dr. G. Bangga and his co-authors are capable of doing it. The second is about the success of the prediction of the center of pressure (Section 3.8). Despite the imprecision on the predictions of CL and CM , it appears that the ratio XP ( = - CM/CL) is well predicted, as though the errors on CL and CM are canceling in the ratio. The improvements proposed so far in the manuscript are not significant to my opinion. I would suggest that the authors look at the model implementation of the stall and flow reattachment delays. For the comparison of the model with experiments based on the first order correction, it would be clearer if the cases of non – stalled conditions are considered, there are no effects of second order for these cases. For the second order model, the main correction to Snel's model proposed by the authors (and Adema et al.) has been to replace the damped oscillator when $d\alpha/dt$ < 0 for a self-excited oscillator of Van-der-Pol type with more damping. The objective has been to capture the oscillatory behavior on the return cycle of the aerodynamic coefficients. However, in Truong's model (see Reference "Modeling aerodynamics for comprehensive analysis of helicopter rotors" by K.V. Truong,42nd European Rotorcraft Forum, Lille, France, September 5-9, 2016 and also published in Aerospace 2017, vol.4, 21), the self-excited oscillator is only replaced by the damped oscillator, when the flow is reattached on the return cycle, i.e. with some lapse of time after the change of sign of $d\alpha/dt$. Under such circumstances, the oscillatory behavior still subsists in the return cycle, albeit with smaller amplitude. This point has been raised also by Dr. X. Munduate while reviewing the article of Adema et al., but these authors have not provided any answer. If the revised model is capable of taking into account the stall delay and the flow reattachment, the authors could solve this issue and improve other predictions, particularly the dependence on the reduced frequency.

---

## Referee Comment (RC2) · Khiem V. Truong (Referee) · 26 Jun 2020

**Comments on the revised manuscript "An improved second order dynamic stall model for wind turbine airfoils"**
**by G. Bangga, T. Lutz and M. Arnold**

Khiem V. Truong (Reviewer 1)
helires@gmail.com

**1. General comments:**

Following my first review, the authors Drs. Galih Bangga, Thorsten Lutz and Matthias Arnold have taken into account some of my critics and have made some improvements in their model predictions. However, the main critics are still unanswered.

I will follow the same steps as in the first review.

**2. Technical comments:**

2.1.  Analysis of the various stall models:

The authors spent a great length of time in analysis of the existing stall models that does not present a great interest for the manuscript objective.

They state in their introduction (lines 45 - 46):

"The main purpose of this paper is to document widely used state-of-the-art dynamic stall models in research and industries".

I would disagree with such  assertion of the above sentence:

(1) Is it "one" of the purposes of the paper and not the "main" purpose?

(2) the claim about the presentation of the state-of-art dynamic models in research and industries is exaggerated:

- The BL model presented in the manuscript corresponds only to the second generation model. There is a third generation model that is in use in UK universities and industries, see Reference "Advancement of aerofoil section dynamic stall synthesis methods for rotor design" by Sheng  et al., The Aeronautical Journal 2012, v. 116 no 1179, pp. 521 – 539. Furthermore, it is not presented under the state-space formulation, different from the formulation of the other models.

- The ONERA model presented corresponds to the ONERA-  Edlin model, the first generation model at ONERA.

(3) The application of the dynamic models is incomplete.

For the BL model, it seems that there is no contribution calculated for the vortex component of the aerodynamic coefficients: see Figure 8. For the ONERA - Edlin model, the authors state that "the original ONERA model cannot predict the drag and moment coefficients" (see Figure 9). An experienced user of the ONERA – Edlin should be able to calculate these coefficients.

2.2. Values of constants used in the IAG model:

- About the value of the critical stall angle $\alpha_{CRIT}$ of airfoils:

To avoid any confusion about the values indicated in Table 5, it is preferable that the authors state that the values of the critical stall angle are related to airfoils with leading edge grit roughness and Reynolds number of 750k in paragraph "2.6" at the position of line 369, the lines 375-381 in the paragraph "3 Results and Discussion" to be moved to there.

- About the values of kS:

It is misleading to call kS as "Strouhal frequency", while it does not have its value (around "0.1"). I suggest to simply call it a constant with value equal to "0.2".

2.3. Sensitivity of the results against applied time step of the solver:

There is no interest in studying the effects of sensitiveness of the integration of models against the step size. The choice of a ODE solver with automatic step variation would do the job better than a ODE solver with fixed time step. It would allow the choice of output at time values requested by the structural code for coupling fluid-structure.

2.4. Quality of the IAG model:

The authors claim the superiority of their model over the others, but their model errors are not quantified. Following my first review, the authors have included the calculation of the L2-norm error, but only for their own model. I would expect such calculation for the Snel, Adema and IAG models in the cases analyzed in Figures 10, 11 and 12. Regarding these cases, I suggest that the values of pitch angles to be used in computation are to be obtained from a fit of experimental values and not from experimental values provided by experimenters. This could lead to a lower L2-norm error and a better graphical visualization of the correlation between experiments and predictions.

**3. Technical corrections:**

The various points raised in the first review have a satisfactory answer, except for the effects of increasing frequency k.

Line 549: "Increasing k above 0.1 reduces the viscous effects and vortex shedding influence".

It is well known that the increase of k leads to a more important variation of the pitching moment (that is not shown in your simulation in Figure 21) and it translates into more structural fatigue.

Additional remarks:

* Figure 21: Effects of k:

  I would suggest studies of k = 0.03, 0.05 and 0.10 instead of high k (0.015 and 0.21): there are few experiments at such high values and the model seems unable to capture effects at high k.

* Line 540: assertion without proof for the BL model.

* Line 542-543: "Despite that, only the Adema model and the present IAG model are able to demonstrate the higher harmonic effects".

The ONERA – Hopf bifurcation does it too!

* Line 546: typo "comparsion".

* Line 548: "without changing the constants"

  Be more precise: by changing only the values of $\alpha$CRIT.

* For references: delete line 623, the reference in line 622 is the same.

**4. Concluding remarks and suggestions for revision:**

  There are some improvements for the model predictions, particularly for the drag coefficients. However, the main critics from the first review are still unanswered satisfactorily.

  I suggest that the authors leave out the BL and ONERA – Edlin models, unless if the authors are willing to spend more effort for studying these models. It appears better to center the effort on the second order model, as stated in the abstract.

---

## Referee Comment (RC3) · Khiem V. Truong (Referee) · 3 Jul 2020

Dear Dr. Galih Bangga,

Your colleagues and you have answered to most of my comments in the second revision of the manuscript. As far as I am concerned as reviewer 1, I'll propose it for publication in WES.

Best regards, K.V. Truong

---

## Referee Comment (RC4) · Gerard Schepers (Referee) · 14 Jul 2020

Thank you very much for this nice article. I think you give a very good overview of various dynamic stall models. You also show a good performance of your new model. Moreover the article is well written and structured.

I went through the revised version which you made after the comments from my fellow reviewer Mr Truong and I donot have much to add.

There are a few relatively minor things which I ask you to consider

- Could you add a section *Recommendation for future work*. This is mainly because I agree to Mr Truongs comments that the airfoils which you consider are thin. Although you reply by saying that these thin airfoils can be found at the tip of HAWT's I think that most of HAWT tip airfoils are 18% or thicker (inboard even very much thicker). Apart from that the Reynolds number is much lower than found on most nowadays wind turbines. A recommendation on a dynamic stall experiment for thicker airfoils as found on modern wind turbines at much higher Reynolds numbers would make sense to me
- I would also appreciate a few words on the limitations of your model:
  - All discussion are 2D. In the very beginning of your article you put some emphasis on 3D effects which are very important for wind turbines indeed but these effects are excluded in the rest of the article.
  - I think the model is tuned for dynamic stall operation at relatively small angles angles of attack only, not for dynamic deep stall which may occur at standstill.
- You often use the word robustness as driver for your work? What do you mean with it? I sometimes interpret it as simplicity, sometimes as general validity or do you mean numerically stable?
- In line 24 you mention that dynamic stall effects can be dangerous. Still dynamic stall generally enhances the aerodynamic damping
- Line 25: Can you be a bit more specific? If the models are working reasonably well why are you trying to improve them. And wrt the very small computational effort: I would write 'without any notable increase in computational effort' or something like that.
- In figure 7 I note that the IAG results are sensitive to time step as well?
- References: I think the list is rather complete and all references seem retraceable. The reference from Ricardo Pereira was a MsC thesis and not a PhD thesis. It may anyhow be better to refer to his article https://repository.tudelft.nl/islandora/object/uuid%3A6e98580d-7f76-493e-a74e-b3f73542b32a/datastream/OBJ/download. Some other TUDelft publications can be found on their repository. You could refer to this repository since this increases accessibility to the background information, an example is https://repository.tudelft.nl/islandora/object/uuid%3Af1ee9368-ca44-47ca-abe2-b816f64a564f
- Notations: I think you manage to give a very good overview of dynamic stall models with consistent notations indeed. These notations are explained on page 34 but you are not 100% complete. For example the reduced frequency k and frequency f are not included. I also note that model constants are excluded from the list. You explain these in the text when they are first introduced but they return at other places and then they are not explained. Please be aware that an ignorant reader might get a bit confused by all these formula. You could help him/her a lot by making a very accurate list of notations including all model constants. Donot forget to add units as well.

**Then a few typos/language issues:**

- Line 59: Mainly
- Line 386 Usually a step of
- Line 396: This sentence which you add as response to Mr Truong's comments does not read well. Maybe you mean:
  *Because dynamic stall models are added to an aerodynamic model based on e.g. BEM, vortex wake or actuator line, which in turn is integrated in a wind turbine solver, the studies are relevant.*
- Line 546: comparison
- Line 551: Again I donot like the sentence which you have added in response to Mr Truong's comments.
  I would write *.... for lift. The opposite is true for the pitching moment*

---

## Editor Comment (EC1) · Alessandro Bianchini (Editor) · 14 Jul 2020

Dear authors, after the positive discussion with Reviewer #1, which made significant improvements in the paper possible, you are encouraged to carefully address also the requests made by Reviewer #2. Looking forward to hearing from you, best regards

---

## Author Comment (AC4) · 17 Jul 2020

Dear Gerard,

first of all we would like to thank you for your comments and suggestions. The positive attitude towards publishing the paper and the constructive feedback are highly appreciated. All remarks given have been considered in the manuscript. The changed texts are indicated by red color in the marked up revised paper. The discussion paper has been revised accordingly.

We hope that the revisions done satisfy your requests.

[Figure]

Kind regards,
Galih Bangga on behalf of the other authors

Please also note the supplement to this comment:
https://wes.copernicus.org/preprints/wes-2020-75/wes-2020-75-AC4-supplement.pdf

---

## Author Comment (AC5) · 17 Jul 2020

Dear Alessandro,

Thanks for the reminder. Our response to Reviewer-2 has been posted online.

Kind regards,
Galih

———————————————

---

## Author Response (AR1)

**Final Response**

Dear Editor and Referees,

We would like to express our gratitude for the time and effort you have spent in reviewing our paper. Your feedback has proven to be crucial for improving the quality of our paper significantly. The paper has been revised appropriately considering all comments, critics and suggestions. The responses to the reviewers' remarks are attached below.

We hope that the revisions done satisfy the reviewers' and editor's requests.

Kind regards,
Galih Bangga on behalf of the other authors

Attachments:
1. Response to Reviewer-1 - Round1
2. Response to Reviewer-1 - Round2
3. Response to Reviewer-1 - Round3
4. Response to Reviewer-2
5. Marked-up manuscript accommodating all review remarks

**Reviewer 1: Dr. Khiem Truong**

Dear Dr. Truong, first of all we would like to thank you for your comments and suggestions. The positive attitude towards publishing the paper and the constructive feedback are highly appreciated. All remarks given have been considered in the manuscript. The changed texts are indicated by red color in the marked up revised paper. The discussion paper has been revised accordingly.

*2.1. Analysis of the various stall models: The authors spent a great length of time in analysis of the existing stall models that does not present a great interest for the manuscript objective. In doing so, the authors have made various mistakes. The Beddoes-Leishman model is not presented under the state-space formulation. Therefore, the sensitivity study of this model against step size of integration cannot be made, as stated in line 393.*

and

*2.3. Sensitivity of the results against applied time step of the solver: The authors use a rudimentary numerical tool for solving the ordinary differential equations (ODE) with fixed time step, there exist more robust ODE solver with automatic step variation. Therefore, the discussion related to the time step size is irrelevant (sec-tion 3.1 and conclusion).*

> Thank you for this highly important comment. Indeed it is correct that the time step assessment for the Leishman-Beddoes model cannot be done since it is not presented in the state-space formulation. Therefore, the analysis for the Leishman-Beddoes model is now removed from the paper. Despite that, the analyses for the other models are still relevant. We do agree that there are more advanced integration approach using variable time step variation. In wind turbine computations, however, a fixed time step approach is often adopted even for high fidelity CFD approaches. Because the proposed dynamic stall model shall be coupled with a separate with turbine load solver, e.g., blade-element momentum, vortex model or actuator line model, therefore the studies are relevant for the community. Thank you for the comment, this motivation is now further clarified in the revised paper as:

> "...numerical uncertainty. The time step assessment for the Leishman-Beddoes model is not included since it is not presented in the state-space formulation. Furthermore, a fixed time step approach is often adopted in wind turbine computations even for high fidelity CFD approaches. Because dynamic stall models shall be coupled with a separate with turbine load solver, e.g., blade-element momentum, vortex model or actuator line model, therefore the studies are relevant. It can be seen clearly..."

*About the ONERA model, they are not aware that it was renamed ONERA-EDLIN ("Equations Différen-tielles Linéaires", meaning in English Linear Differential Equations), to distinguish it for the newer model ONERA-BH ("Bifurcation de Hopf", renamed later by his author as ONERA Hopf Bifurcation model). It is usual for researchers in the field of wind turbines to continue to call it with such name; so, this mistake is not serious. The critical error of the authors is to not consider the stall delay in the ONERA-EDLIN model. Without the account of stall delay, this model leads to predictions of the lift coefficient with large discrepancies in correlation with experiments as shown in Figure 9 p.20*

> Thank you for this information. Regarding the ONERA model, we are aware that we are not using the latest updated version of the model as pointed out by the reviewer. The main objective of the paper is to assess the IAG model for various airfoils and flow conditions. Therefore we decided to use the basic ONERA model equations, not the updated version, because it serves only for a short comparison - not for evaluating the ONERA model itself. We followed the model presented by Holierhoek et al in their paper [1]. Despite that, according to your recommendation, we updated the

ONERA model to account for the stall delay effect. Indeed the prediction accuracy is improved. Therefore, we have now added the information you provided and the corrections into our paper as:

"... As shown by the ONERA-EDLIN (Equations Différentielles Linéaires) model [2], the standard formulation without considering the stall delay effect tends to underpredict the lift force above stall. In order to account for the stall delay effect as in [3], the value of $\Delta C_{L_n}^{INV}$ is kept constant for a specified time after stall ($s_{end} - s_{stall} = 8$). A similar procedure was adopted in [4] for the stall delay model..."

and the prediction is improved as:

[Figure]

Figure 1: Dynamic force reconstruction using the ONERA model in comparison with the measurement data [5] for $\Delta t = T/1440$. S801 airfoil, $k = 0.073$, $\overline{\alpha} = 20°$, $\Delta\alpha = 10°$. Left: without stall delay, right: with stall delay.

*2.2. Values of constants used in the IAG model:There are two types of constants used for the IAG model that are ill chosen, the critical stall angle and the value of the Strouhal number. The critical stall angle $\alpha CRIT$ of airfoils is one key parameter for the stall model. The authors choose this value based on the position of the break of the pitching moment coefficient and the position of the important increase of the drag coefficient. This is nota good choice, as pointed by Sheng et al. in their conclusions (Reference cited on line 598), the best choice is the incidence angle at the maximum chord force coefficient.Led by such bad criteria of defining $\alpha CRIT$ , the authors found very small values for theairfoils S801, S809 and S814 : 15.1o, 14.1o and 10o respectively, instead of the values of 17.6o , 19.2o and 13.9o found by Sheng et al. (Reference Sheng W., GalbraithR.A.McD. and Coton F.N., "Applications of low-speed dynamic-stall model to the NREL airfoils", Journal of Solar Energy Engineering, 2010, vol. 132, pp. 011006-1:011006-8). The increase of the value of $\alpha CRIT$ would allow a better correlation of their model predictions with experiments, as shown in Figures 10-12, and following*

Thank you for the information and comments. The critical angle we selected is actually consistent with the one used by Sheng et al. The angles presented by Sheng et al are larger than the one used in the present work because they are taken from polars with natural transition (see Figure 2 for S809 airfoil below), in contrast we employed the polar data with transition trip as stated in the beginning of Section 3 as "All selected test cases are for the airfoils employed with a leading edge grit (turbulator) to enable the "soiled" effects on a wind turbine blade at a Reynolds number of around 750K.". To avoid confusion, we revised the sentence as:

"All selected test cases are for the airfoils employed with a leading edge grit (turbulator) to enable the "soiled" effects on a wind turbine blade at a Reynolds number of around 750K. Note that these polar data are different with the one used for example by Sheng et al. [6] where the natural transition cases were taken. Therefore, the critical angles of attack are also different.

[Figure]

Figure 2: Determination of $\alpha^{CRIT}$. Note that the scale of $y$-axis for each force component is plotted independently for clarity.

We also would like to inform you that we made a typo regarding the critical angle of attack for the NACA 4415 airfoil. Now it has been corrected in the revised paper.

*Value of the Strouhal number S: the authors following Adema et al. use the value of S = 0.2; they should notice from various references that S is in the range of [0.06,0.13](see for example "Spectral analysis of New MEXICO standstill measurements to investigate vortex shedding in deep stall" by Khan M.A., Ferreira C.S., Schepers G.J.and SØrensen N.N., Wind Energy, 2019, pp.1-14). When S decreases, the predicted distance between two consecutive extremum (maximum for the lift and drag coefficients, minimum for the pitching moment coefficient) of the aerodynamic coefficients increases. The correlation between model predictions and experiments would be improved*

Thank you for your very important recommendation. The word "Strouhal number" in the formulation is actually not the real Strouhal number itself because the effect is controlled by the applied constants in the ODE. As for example, we tried changing the value to be smaller, as a result the accuracy degrades without calibrating the other constants. By reducing the value, one can see in Figure 3 that the higher harmonic effects disappear accordingly, which is not preferable. Therefore, a value of 0.2 is taken for the standard airfoil analyses.

[Figure]

Figure 3: Effects of specified "Strouhal number" on predicted $C_L$ (top) and $C_M$ (bottom). From left to right: St = 0.06, St = 0.13, St = 0.2. Red curves are prediction results, black curves are experiment. NACA4415 airfoil, $k = 0.073$, $\overline{\alpha} = 20°$, $\Delta\alpha = 10°$.

*2.4. Quality of the IAG model: The authors claim the superiority of their model over the others, but their model errors are not quantified. Since the study of Holierhoek et al. (cited in line 574), practically all the publications on hysteresis loops in stalled conditions of airfoils provide the value sof the error L2-norm, see for instance the publication of Adema et al. I would consider that the predictions for the lift coefficient are reasonable. However, the predictions of the drag coefficient are overestimated and this would lead to under-prediction of the power coefficient CQ. The predictions of the pitching moment coefficient are not right in some cases. For instance for the airfoil S801 in Figure 17, the predictions show clockwise hysteresis sub-loop that correspond to negative aerodynamic damping,while the experiments show anti-clockwise sub-loop leading to positive aerodynamic damping.*

and

*For the comparison of the model with experiments based on the first order correction, it would be clearer if the cases of non – stalled conditions are considered, there are no effects of second order for these cases.*

Thank you for the remarks and suggestions. We have now added a section assessing the $L_2$-norm of errors for cases involving different airfoils. In order to limit the number of pages and since the attached flow regime is not our main focus, a dedicated analysis for the attached flow regime will not be directly presented in the paper. Despite that, we do agree with your suggestion. Therefore, to accommodate this aspect, the $L_2$-norm of errors are quantified for cases involving different airfoils; both under attached and deep stall conditions. Indeed the errors for the deep stall cases are larger, but still at reasonable values. In fact, most flow cases considered in our studies are for the deep stall conditions.

*2.5. The study of various airfoils: It would be interesting that the model predictions could show some distinctive features associated with the thickness for the airfoils studied, ranging from thin (S801) to thick airfoil (S814). Thin airfoils are characterized by leading-edge stall, whereas thick airfoils by trailing-edge stall. The choice of the airfoil S801 by the authors of the submitted paper for extensive studies is unfortunate, because it is a thin airfoil of thickness 13.5%,and wind turbine blades have usually thickness larger than 15%.*

We do agree that the S801 airfoil is relatively thin compared to the usual wind turbine airfoils.

[Figure]

Figure 4: Quantified $L_2$ norm of error with respect to the measurement data for four airfoils. Top: attached flow case ( $k = 0.073$, $\overline{\alpha} = 8°$, $\Delta\alpha = 5.5°$), bottom: deep stall case ( $k = 0.073$, $\overline{\alpha} = 20°$, $\Delta\alpha = 10°$)

Despite that, thin airfoils are still practically used for wind turbines especially near the tip regimes. In vertical axis wind turbines, where dynamic stall plays a major role, the airfoil is often relatively thin and the present studies should be of interest. we want to evaluate the model for various stall characteristics as the main purpose of the paper. More importantly, the practical use of the model shall not be limited only to wind energy applications.

*Line 4: the sentence "many flow parameters" is not clear.Line 538:*

Thank you for the correction. It is now corrected as:

"....Comprehensive investigations and tests are performed at various flow conditions..."

*"Increasing k above 0.1 leads to an increased flow stability": this is incorrect.*

Thank you for the correction. The phrase has now been removed and corrected as:

"Increasing $k$ above 0.1 reduces the viscous effects and vortex shedding influence."

*Lines 539-540: the assertion is incorrect. For a large angle variation, the variation of the pitching moment coefficient is more important (see Figure 24) and this could lead to more structural damage to the blades.*

Thank you for the correction. The sentence has now been revised as:

"When the airfoil operates at a high $\overline{\alpha}$ within the stall regime, a small $\Delta\alpha$ leads to increased vibrations for lift, but contrary for the pitching moment.

*Imprecision for the section References: - Lines 555, 558, 566, 570, 577, 585, 598 and 604. Inconsistencies for Lines 568 and 574*

Thank you for the comments. The references and inconsistencies have now been corrected:

*Concluding remarks and suggestions for revision: Though the submitted paper is marred with errors, there are two positive aspects. The first one is about the objective of examining Snel's model for various flow conditions and airfoils. This stall model has been around 1997 and no exhaustive evaluation has been made at my knowledge. I feel that Dr. G. Bangga and his co-authors are capable of doing it. The second is about the success of the prediction of the center of pressure(Section 3.8). Despite the imprecision on the predictions of CL and CM , it appears that the ratio XP ( = - CM/CL) is well predicted, as though the errors on CL and CM are canceling in the ratio.*

Thank you for the remarks given.

*For the second order model, the main correction to Snel's model proposed by the authors (and Adema et al.) has been to replace the damped oscillator when dα/dt < 0 for a self-excited oscillator of Van-der-Pol type with more damping. The objective has been to capture the oscillatory behavior on the return cycle of the aerodynamic coefficients. However, in Truong's model (see Reference "Modeling aerodynamics for comprehensive analysis of helicopter rotors" by K.V. Truong,42nd European Rotorcraft Forum, Lille, France, September 5-9, 2016 and also published in Aerospace 2017,vol.4, 21), the self-excited oscillator is only replaced by the damped oscillator, when the flow is reattached on the return cycle, i.e. with some lapse of time after the change of sign of dα/dt. Under such circumstances, the oscillatory behavior still subsists inthe return cycle, albeit with smaller amplitude. This point has been raised also by Dr.X. Munduate while reviewing the article of Adema et al., but these authors have not provided any answer. If the revised model is capable of taking into account the stall delay and the flow reattachment, the authors could solve this issue and improve other predictions, particularly the dependence on the reduced frequency.*

Many thanks for the suggestions. This last comment is extremely helpful for us to improve our model. Using your remarks as the starting points, we updated our model accommodating several aspects. First, we evaluated the location where the drag force starts to increase and tried to relate the position with a weighted separation point $\zeta$. By doing so, one can set a better drag limiter than our previous definition. This is revised in the paper as:

".....If one uses this formulation directly, at some point drag still becomes lower than the static drag value by a significant amount. By evaluating the experimental data for several airfoils and various flow conditions, this is not physical at small angles of attack especially in the downstroke regime, where it usually just returns to the static value. In fact, those experimental data infer that strong drag hysteresis occurs only at high angles of attack beyond stall. Similarly, in the upstroke regime the drag value increases only slightly (approximately only 20%). In Figure 5, one can see that drag hysteresis occurs when

$$\zeta = \frac{1}{\pi} \frac{dC_N}{d\alpha} \left( \frac{1 + \sqrt{f_n}}{2} \right)^2 \lesssim 0.76. \tag{1}$$

Based on these observations, a simple drag limiting factor is adopted when $\zeta_n \geq 0.76$ as:

$$C_{D_n}^D = \begin{cases} 1.2 C_{D_n}^{VISC}; & \text{if} \quad C_{D_n}^D > 1.2 C_{D_n}^{VISC} \quad \text{and} \quad \left( C_{N_n}^P - C_{N_{n-1}}^P \right) \geq 0.0 \\ C_{D_n}^{VISC}; & \text{if} \quad \left( C_{N_n}^P - C_{N_{n-1}}^P \right) < 0.0 \\ C_{D_n}^D; & \text{otherwise} \end{cases} \tag{2}$$

Note that for the purpose of numerical implementation, it is always recommended in practice to adopt relaxation to avoid any discontinuity which may present in the above formulation. The effects of these modifications are displayed in Figure 6.

[Figure]

(a)  (b)  (c)  (d)

Figure 5: Relation between drag hysteresis in the stall regime with weighted separation parameter $\zeta$ for four airfoils. From left to right: S801 (13.5%), NACA4415 (15%), S809 (21%) and S814 (24%).

[Figure]

(a)  (b)  (c)

Figure 6: Drag reconstruction in comparison with the experimental data for S801 airfoil [5] applying: (a) Equation (19), (b) Equation (70) and (c) Equations (70) + (72).

When the second order term is included in the formulation, one obtains a much better agreement than the previous definition as shown in Figure 7:

[Figure]

Figure 7: Drag reconstruction by the IAG model in comparison with the measurement data [5, 7–9] for $\Delta t = T/1440$ using the actual angle of attack in the experimental campaign for different airfoils. From left to right: S801 (13.5%), NACA4415 (15%), S809 (21%) and S814 (24%). $k = 0.073$, $\overline{\alpha} = 20°$, $\Delta\alpha = 10°$. Top: previous model, bottom: revised model.

Second, by considering your remarks of the second order term regarding the oscillation characteristics of the polar in the downstroke regime as the flow is reattached, we added an additional term when the angle is smaller than the critical angle of attack. This marks the regime where the flow starts to reattach on the airfoil surface.

"...The idea for the downstroke damping as in Equation (66) is adopted in the present model, the following form and constants are used:

$$Kf_{21_n} = \begin{cases} 150k_s[-0.01(\Delta C_{N_n}^{INV} - 0.5) + 2(\Delta C_{N_n}^{D2})^2]; & \text{if} \quad \dot{\alpha_n} > 0 \\ 30k_s[-0.01(\Delta C_{N_n}^{INV} - 0.5) + 14(\Delta C_{N_n}^{D2})^2]; & \text{if} \quad \dot{\alpha_n} \leq 0 \quad \text{and} \quad \alpha_n \geq \alpha_n^{CRIT} \\ 0.2k_s; & \text{if} \quad \dot{\alpha_n} \leq 0 \quad \text{and} \quad \alpha_n < \alpha_n^{CRIT} \end{cases} \quad (3)$$

Note again that $\tau$ is not present in the above equation. The original formulation in Equation (66) replaces the damped oscillator when $\dot{\alpha_n} \leq 0$ for a self-excited oscillator of Van-der-Pol type with more damping. This is in contrast with the implementation done in [10,11], where the self-excited oscillator is only replaced by the damped oscillator, when the flow is reattached on the return cycle. Under such circumstances, the oscillatory behavior still subsists in the return cycle, albeit with smaller amplitude. To accommodate this aspect, the last term of Equation (3) is applied when the angle is smaller than $\alpha_n^{CRIT}$. As for the forcing term, the original form of the Snel model [12] is adopted...."

**Reviewer 1: Dr. Khiem Truong**

Dear Dr. Truong, first of all we would like to thank you for your updated comments and suggestions. The positive attitude towards publishing the paper and the constructive feedback are highly appreciated. All remarks given have been considered in the manuscript. The changed texts are indicated by red color in the marked up revised paper. The discussion paper has been revised accordingly.

*2.1. Analysis of the various stall models: The authors spent a great length of time in analysis of the existing stall models that does not present a great interest for the manuscript objective. They state in their introduction (lines 45 - 46):"The main purpose of this paper is to document widely used state-of-the-art dynamic stall models in research and industries".I would disagree with such assertion of the above sentence:(1) Is it "one" of the purposes of the paper and not the "main" purpose? (2) the claim about the presentation of the state-of-art dynamic models in research and industries is exagerated:*

Thank you for your comment. The statement has now been revised as:

"...Therefore, one major key for a model to be used in industrial applications is robustness of the model itself. One of the purposes of this paper is to document widely used dynamic stall models in research and industries. These include the first order LB model and the second order Snel model. A very recently improved Snel model according to Adema [1] will also be evaluated. The mathematical formulations of these models..."

*The application of the dynamic models is incomplete*

and

*I suggest that the authors leave out the BL and ONERA – Edlin models, unless if the authors are willing to spend more effort for studying these models. It appears better to center the effort on the second order model, as stated in the abstract.*

Thank you for this important suggestion. We have now removed the assessments of the Leishman-Beddoes and ONERA models. Only the second order models are retained in the revised paper. Consequently, the description of the ONERA model in Section 2.2 is removed as well. However, the description of the Leishman-Beddoes model in Section 2.1 is retained because it becomes the founding basis for the IAG model.

*2.2. Values of constants used in the IAG model: To avoid any confusion about the values indicated in Table 5, it is preferable that the authors state that the values of the critical stall angle are related to airfoils with leading edge grit roughness and Reynolds number of 750k in paragraph "2.6" at the position of line 369, the lines 375-381 in the paragraph "3 Results and Discussion" to be moved to there.*

Thank you for the suggestion. The sentences have been moved to the requested location.

*Value of the Strouhal number S: It is misleading to call kS as "Strouhal frequency", while it does not have its value (around "0.1"). Isuggest to simply call it a constant with value equal to "0.2"*

Thank you for your this recommendation. We do agree with you in this regard. The terms Strouhal number has been revised as:

"....Variable $k_s$ is a constant with a typical value of 0.2..."

*2.3. Sensitivity of the results against applied time step of the solver: There is no interest in studying the effects of sensitiveness of the integration of models against the step size. The choice of a ODE solver with automatic step variation would do the job better than a ODE solver with fixed time step. It would allow the choice of output at time values requested by the structural code for coupling fluid-structure.*

and

*Line 540: assertion without proof for the BL model.*

Thank you for your comment. The time step size assessment has been removed from the paper.

*2.4. Quality of the IAG model: The authors claim the superiority of their model over the others, but their model errors are not quantified. Following my first review, the authors have included the calculation of the L2-norm error, but only for their own model. I would expect such calculation for the Snel, Adema and IAG models in the cases analyzed in Figures 10, 11 and 12. Regarding these cases, I suggest that the values of pitch angles to be used in computation are to be obtained from a fit of experimental values and not from experimental values provided by experimenters. This could lead to a lower L2-norm error and a better graphical visualization of the correlation between experiments and predictions.*

Thank you for the remarks and suggestions. We have now added a section assessing the $L_2$-norm of errors for cases involving different airfoils also for the other models. The angle of attack was obtained by interpolating the actual measured angle using a third order spline interpolation, the same approach adopted throughout the paper for the IAG model. This is presented as:

"....Holierhoek et al . [2] introduced a way for quantifying the absolute error between the experimental data and modeled lift coefficient. The general formulation reads:

$$L_2^\phi = \sqrt{\frac{1}{N} \sum_i^N \left( \phi_i^{mod} - \phi_i^{exp} \right)^2} \tag{1}$$

with $\phi$ being the variable of interest, $i$ is the current sample and $N$ is the total number of sample. In their paper, however, only lift was considered. Here all three force components will be shown for four different airfoils. Figure 1 displays the quantified error for two different flow category, attached and deep stall. The timeseries of the angle of attack was obtained from the measured data by applying a third-order cubic-spline interpolation in between each available point. One can see that generally the attached flow case is predicted very well, while the error increases as the flow condition becomes more complicated. Interestingly, especially for lift, it seems that the error reduces with increasing airfoil thickness. The reason for the larger error obtained for the thinner airfoil is attributed to the complex characteristics of the leading edge stall, causing severe load variations especially with increasing angle of attack. Thus, it makes the prediction more challenging. Furthermore, the quantification of the error was also performed on two other dynamic stall models, Snel and Adema-Snel. The same approach for the angle of attack signal was applied. One can see that the IAG model shows its improved prediction especially for the deep stall case for all three force components...."

[Figure]

Figure 1: Quantified $L_2$ norm of error with respect to the measurement data for four airfoils. Top: attached flow case ( $k = 0.073$, $\overline{\alpha} = 8°$, $\Delta\alpha = 5.5°$), bottom: deep stall case ( $k = 0.073$, $\overline{\alpha} = 20°$, $\Delta\alpha = 10°$)

*I would suggest studies of k = 0.03, 0.05 and 0.10 instead of high k (0.015 and 0.21).*

and

*Line 549: "Increasing k above 0.1 reduces the viscous effects and vortex shedding influence". It is well known that the increase of k leads to a more important variation of the pitching moment.*

Thank you very much for this very important remark. We think there is some misunderstanding concerning our previous statement. To avoid any confusion, we rewrite the analyses of the reduced frequency in the revised paper. Indeed we agree with you that the vibrations become more violent with increasing $k$, which can be dangerous for the airfoil structure. The analyses are now rewritten as:

"....The gradient of the lift polar in the upstroke and downstroke phase is also increasing as well. These characteristics are present in both experimental data and predictions delivered by the IAG model. A similar behavior is also displayed in drag and pitching moment in Figures 16 and 17, respectively. It is obvious that stall occurs much earlier for a smaller $k$ value. One can see that the maximum amplitude of all three force components increases with increasing $k$. This can be dangerous for the structural stability, since the amplitude determines the fatigue loads.

To better investigate the effects of $k$, the IAG model is used to reconstruct the dynamic polar data at various $k$ by applying an idealized sinusoidal motion as presented in Figure 2. Only the last DS cycle is shown for clarity of the observation. While the maximum amplitude of all three force components at low frequency domains increases with increasing $k$ (blue and green markers), the amplitudes for all three forces at high frequency domains show different characteristics as shown in the Fourier transformation in Figure 3, albeit with much smaller values. The higher harmonic amplitudes are attributed to flow separation effects, while for low frequency domains are driven by the pitching motion (i.e., external unsteadiness or inflow)....."

[Figure]

Figure 2: Effects of $k$ on the aerodynamic response by the IAG model for $\Delta t = T/1440$. S801 airfoil, $\overline{\alpha} = 20°$, $\Delta \alpha = 10°$. Top: polar, bottom: timeseries.

[Figure]

Figure 3: Fourier transformation of the predicted forces presented in Figure 2. $f_s = f/f_0$ with $f0$ being the pitching frequency.

*Line 542-543: "Despite that, only the Adema model and the present IAG model are able todemonstrate the higher harmonic effects".The ONERA – Hopf bifurcation does it too!*

Thank you for the comment. This sentence applies only to the models investigated in the present paper. To avoid misunderstanding, it is now rephrased as:

"The general characteristics of the polar data can be predicted by all investigated dynamic stall models. Despite that, only the Adema model and the present IAG model are able to demonstrate the higher harmonic effects among the three investigated models."

*Line 546: typo "comparsion".*

Corrected.

*Line 548: "without changing the constants" Be more precise: by changing only the values of $\alpha CRIT$.*

Corrected as:

"The dynamic forces reconstructed by the IAG model are in a sound agreement with the experimental data under various flow conditions by variation of $\overline{\alpha}$, $k$, $\Delta\alpha$ and for four different airfoils by changing only the values of the critical angle of attack.

*For references: delete line 623, the reference in line 622 is the same*

Done.

The above discussions sum up the corrections performed in the updated paper. We hope that the revisions done satisfy the reviewer's requests.

**Reviewer 1: Dr. Khiem Truong**

Dear Dr. Galih Bangga,

Your colleagues and you have answered to most of my comments in the second re-vision of the manuscript. As far as I am concerned as reviewer 1, I'll propose it for publication in WES.

Best regards,
K.V. Truong
* * *
Dear Dr. Truong,

at last but not least, in this final reply comment we would like to thank you for providing us constructive remarks, critics and suggestions. Our paper has been improved considerably by following your recommendations.

Kind regards,
Galih Bangga on behalf of other authors

**Reviewer 2: Dr. Gerard Schepers**

Thank you very much for this nice article. I think you give a very good overview of various dynamic stall models. You also show a good performance of your new model. Moreover the article is well written and structured. I went through the revised version which you made after the comments from my fellow reviewer Mr Truong and I do not have much to add. There are a few relatively minor things which I ask you to consider.
* * *
Dear Gerard,

first of all we would like to thank you for your comments and suggestions. The positive attitude towards publishing the paper and the constructive feedback are highly appreciated. All remarks given have been considered in the manuscript. The changed texts are indicated by red color in the marked up revised paper. The discussion paper has been revised accordingly.

*Could you add a section Recommendation for future work. This is mainly because I agree to Mr Truongs comments that the airfoils which you consider are thin. Although you reply by saying that these thin airfoils can be found at the tip of HAWT's I think that most of HAWT tip airfoils are 18% or thicker (inboard even very much thicker). Apart from that the Reynolds number is much lower than found on most nowadays wind turbines. A recommendation on a dynamic stall experiment for thicker airfoils as found on modern wind turbines at much higher Reynolds numbers would make sense to me*

and

*I would also appreciate a few words on the limitations of your model: 1. All discussion are 2D. In the very beginning of your article you put some emphasis on 3D effects which are very important for wind turbines indeed but these effects are excluded in the rest of the article. 2. I think the model is tuned for dynamic stall operation at relatively small angles angles of attack only, not for dynamic deep stall which may occur at standstill.*

Thank you for your suggestion. We do agree with you. Adding a section about future research studies will be good to encourage further investigations and assessment of the model. Furthermore, you are right that 3D effects could play a role especially at large angles of attack. This will be also suggested as future work. These comments are now blended in the last section as:

"...

**5 Recommendations for Future Work**

The present paper evaluates the newly developed IAG model under various flow conditions for four different airfoils. The following aspects are encouraged for future work:

- In the present studies, the assessment was mainly carried out for the S801 airfoil having a relative thickness of 13.5%. This airfoil is mainly characterized by leading edge separation, which is very challenging for validating the accuracy of a dynamic stall model. However, typical modern wind turbine blades usually employ airfoils with no less than 18% relative thickness and at much higher Reynolds number. Therefore, future investigations shall be done for thicker airfoils at various flow conditions as well.

- The above statement is also true for the current available experimental data. Therefore, experiments on dynamic stall for thick airfoils at much higher Reynolds number are encouraged.

- Three-dimensional effects (Himmelskamp or tip loss effects) for a rotating blade can alter the loads significantly even under a steady inflow condition. Further consideration and examination of the model under this condition shall be done.

- Further tests and re-calibration of the model for deep-stall conditions at extremely large angles of attack are encouraged, which can be relevant for a turbine in stand still.

..."

*You often use the word robustness as driver for your work? What do you mean with it? I sometimes interpret it as simplicity, sometimes as general validity or do you mean numerically stable?*

Thank you for your question. To avoid misconception, the sentence is further clarified in introduction as:

"...in industrial applications is robustness of the model itself, i.e., the model is easy to apply with small number of well defined user parameters. One of the purposes of this paper is..."

*In line 24 you mention that dynamic stall effects can be dangerous. Still dynamic stall generally enhances the aerodynamic damping*

We do agree with this statement. However, dynamic stall causes enhanced peak-to-peak loading that increases the fatigue stress. For completeness, your comment is further added in the text as:

"...lift coefficient ($C_L$) and can be dangerous for the blade structure itself, although dynamic stall also generally enhances aerodynamic damping..."

*Line 25: Can you be a bit more specific? If the models are working reasonably well why are you trying to improve them. And wrt the very small computational effort: I would write "without any notable increase in computational effort" or something like that.*

Thank you for the comment, the sentence is revised as:

"...To model the behavior of the airfoil under these situations, semi-empirical models can be used. The models are known to produce reasonable results without any notable increase in computational effort. Despite that, these models usually cannot reproduce higher harmonics of the load fluctuations. Furthermore, the applied constants shall be adjusted according to the flow conditions and airfoils...."

*In figure 7 I note that the IAG results are sensitive to time step as well?*

Thank you for the comment. Although not much, the IAG model indeed shows some dependency upon the applied time step size. However, as per request from Reviewer-1 (Dr. Truong), this section has been removed from the paper.

*References: I think the list is rather complete and all references seem retraceable. The reference from Ricardo Pereira was a MsC thesis and not a PhD thesis. It may anyhow be better to refer to his article https://repository.tudelft.nl/islandora/object/uuid%3A6e98580d-7f76-493e-a74e-b3f73542b32a/datastream/OBJ/download. Some other TUDelft publications can be found on their repository. You could refer to this repository since this increases accessibility to the background information, an example is*

*https://repository.tudelft.nl/islandora/object/uuid%3Af1ee9368-ca44-47ca-abe2-b816f64a564f*

Thank you for the recommendation. The reference has been updated and the repository has been added for the publications from Delft.

*Notations: I think you manage to give a very good overview of dynamic stall models with consistent notations indeed. These notations are explained on page 34 but you are not 100% complete. For example the reduced frequency k and frequency f are not included. I also note that model constants are excluded from the list. You explain these in the text when they are first introduced but they return at other places and then they are not explained. Please be aware that an ignorant reader might get a bit confused by all these formula. You could help him/her a lot by making a very accurate list of notations including all model constants. Donot forget to add units as well.*

Thank you for the comment. The list of notations has been updated.

———————————————————————

Then a few typos/language issues:

*Line 59: Mainly*

Corrected.

*Line 386 Usually a step of*

As per request from Reviewer-1 (Dr. Truong), this section has been removed from the paper.

*Line 396: This sentence which you add as response to Mr Truong's comments does not read well. Maybe you mean: Because dynamic stall models are added to an aerodynamic model based on e.g. BEM, vortex wake or actuator line, which in turn is integrated in a wind turbine solver, the studies are relevant.*

As per request from Reviewer-1 (Dr. Truong), this section has been removed from the paper.

*Line 546: comparison.*

Corrected.

*I would write .... for lift. The opposite is true for the pitching moment.*

Corrected.

———————————————————————

The above discussions sum up the corrections performed in the updated paper. We hope that the revisions done satisfy the reviewer's requests.

Kind regards,
Galih Bangga on behalf of the other authors

[revised manuscript text omitted]